# Experimental study of transonic flow over a wind turbine airfoil

Abhyuday Aditya[1], Maria Cristina Vitulano[1], Delphine De Tavernier[1], Ferdinand Schrijer[1], Bas van Oudheusden[1], and Dominic von Terzi[1]

[1]Faculty of Aerospace Engineering, Delft University of Technology, 2629HS Delft, The Netherlands

**Correspondence:** Abhyuday Aditya (a.aditya@tudelft.nl)

**Abstract.** For the largest wind turbines currently being designed, operation close to cut-out conditions can lead to the tip airfoil experiencing transonic flow conditions. To date, this phenomenon has been explored primarily through numerical simulations, but modelling uncertainties limit the reliability of these predictions. In response to this challenge, our study marks the first experimental investigation of a wind turbine airfoil under transonic conditions, for which we selected the FFA-W3-211 airfoil. Measurements were carried out in the high-subsonic range (Mach 0.5 and 0.6), utilizing schlieren visualization and Particle Image Velocimetry to characterize the airfoil across a range of angles of attack expected to be close to the boundary of transonic flow occurrence. Unsteady shock wave formation was observed for the higher Mach number, with the shock oscillation range increasing with steeper angles of attack. In addition, it was confirmed that the presence of a local supersonic flow region does not necessarily result in a shock wave. For cases with shock waves and trailing-edge separation, a buffet cycle was identified that is similar to, but distinct from, those seen in aviation applications. Our findings highlight the need for unsteady analyses even in steady operating conditions and call for dedicated research on wind turbine tip airfoils in transonic flow.

## 1 Introduction

To help meet the growing global demand for energy in an environmentally sustainable manner, wind turbines have been steadily increasing in size, allowing them to capture a greater portion of the wind energy potential (Mehta et al., 2024a). This trend is driven by the need to optimize power generation to meet market demands (Mehta et al., 2024b). As a result, the next generation of offshore wind turbines is expected to have rotor diameters greater than 250 m. Such a dramatic increase in scale also introduces unprecedented aerodynamic challenges. At the tips of such gigantic rotors, the resulting flow speed is around 100 m/s, which translates to a Mach number of $\sim 0.3$. At such inflow Mach numbers, flow compressibility cannot be assumed to be negligible.

For instance, the latest reference wind turbine (RWT), the IEA 22 MW RWT, is designed with a rotor diameter of 280 m and can operate with blade tip speeds of up to 105 m/s (Zahle et al., 2024). At a cut-out wind speed of approximately 25 m/s, the resulting relative flow at the tip airfoil exceeds a Mach number of 0.3. Moreover, the high camber of typical wind turbine airfoils leads to rapid flow acceleration over the airfoil, with the possibility of inducing local pockets of supersonic flow, i.e., transonic flow.

Compressibility and transonic flow introduce complex flow phenomena such as shock wave formation, which can reduce aerodynamic efficiency and impose additional structural loads. Despite their importance, high-speed compressibility effects

in wind turbine operations remain relatively underexplored. One of the first investigations into these dynamics was by Wood (1997), who examined small horizontal-axis wind turbines with NACA 0012 airfoils and proposed using shock-induced separation at blade tips for overspeed protection. However, symmetric airfoils like the NACA 0012 are not representative of modern utility-scale turbines, limiting the applicability of these findings.

A subsequent study by Hossain et al. (2013) analyzed shock propagation over the NREL Phase VI S809 airfoil using two-dimensional Reynolds-Averaged Navier-Stokes (RANS) simulations at various angles of attack. Although informative, this work focused on a free-stream Mach number of 0.8, which is well beyond the normal conditions anticipated in current or foreseeable wind turbine designs, making it less relevant to large-scale applications.

More recently, De Tavernier and von Terzi (2022) explored the potential for transonic flow occurrence over the IEA 15 MW reference wind turbine (RWT) (Gaertner et al., 2020). Their study emphasized how offshore operating conditions, such as high atmospheric turbulence, blade aeroelastic effects, and floating platform motions, can increase the instantaneous wind speeds experienced by blade tips. Using XFoil and OpenFAST, they showed that near rated power and close to cut-out wind conditions, the tip airfoil could encounter large negative angles of attack (AoAs) due to pitch control actions and unsteady inflow. When combined with elevated tip Mach numbers, these factors led to intermittent pockets of local supersonic flow at the blade tip. However, XFoil cannot predict whether local supersonic flow results in shock waves.

Building on this work, Vitulano et al. (2025a) conducted a more detailed investigation of transonic effects on the FFA-W3-211 airfoil using unsteady RANS (URANS) simulations, which are better equipped than XFoil to capture complex flow physics such as shock waves. Their results confirmed that local supersonic flow regions do indeed coalesce into shock waves over the FFA-W3-211 airfoil (used at the IEA 15 MW RWT tip) under similar operating conditions (inflow Mach number and angle of attack) as predicted by De Tavernier and von Terzi (2022). Their investigation also revealed that apart from the inflow Mach number and angle of attack, the onset of transonic flow and shock waves was also heavily dependent on the Reynolds number, especially at steeper inclinations.

The phenomenon of shock waves in transonic flow is particularly concerning because they can degrade aerodynamic performance and induce strong flow unsteadiness. When coupled with flow separation, self-sustained shock oscillations known as transonic buffet can occur. While the underlying mechanisms of transonic buffet remain a topic of open research (Lee, 2001; Giannelis et al., 2017), its consequences are well established: periodic load fluctuations, vibrations, and resonance risks that threaten structural integrity. The primary parameters that determine the onset of transonic buffet on a supercritical airfoil are inflow Mach number and the angle of attack, as demonstrated by numerous experimental studies (Jacquin et al., 2009; Accorinti et al., 2022) and URANS simulations (Giannelis et al., 2018). Furthermore, significant asymmetries in the flow-field exist during a transonic buffet cycle. As investigated by D'Aguanno et al. (2021), a larger separation region is observed when the shock moves upstream compared to downstream. These findings highlight the strong inherent unsteadiness of transonic buffet, even under steady inflow and static airfoil conditions.

As illustrated in Fig. 1, wind turbine tip airfoils, like the FFA-W3-211, differ markedly from supercritical airfoils used in aviation, e.g., the OAT 15A, featuring a higher thickness to chord ratio, greater camber, and distinct operating conditions. Notably, the steep negative angles of attack encountered by wind turbine tips in above rated wind speeds are not typical of the

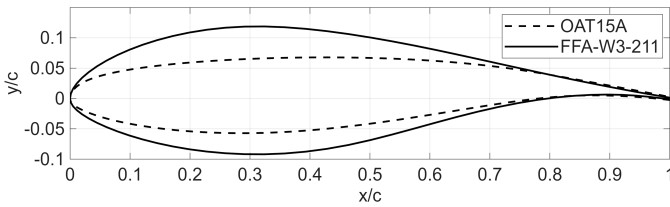

**Figure 1.** Comparison of the OAT 15A supercritical airfoil (dotted line) and the FFA-W3-211 wind turbine airfoil (solid line).

widely researched transonic buffet that occurs over supercritical airfoils. This emphasizes that transonic buffet on wind turbine airfoils could be strikingly distinctive and is, therefore, in need of dedicated research.

For investigating complex turbulent flows, URANS is a popular technique as it offers a good compromise in terms of fidelity and computational expense between higher-fidelity but extremely resource-intensive techniques – like Large Eddy Simulation (LES) or Direct Numerical Simulations (DNS) – and low-fidelity models like XFoil that do not capture shock waves at all. However, URANS results rely heavily on turbulence modelling assumptions and may struggle to capture the correct physical behaviour, especially for highly nonlinear phenomena such as transonic buffet (Illi et al., 2012). This highlights the need to obtain experimental results that can serve for URANS validation.

The present study addresses this research gap by experimentally investigating transonic flow occurrence over the FFA-W3-211 airfoil, which is used at the blade tips of both the IEA 15 MW and 22 MW RWTs. Through detailed experimental characterization, this research aims to provide crucial insights into the transonic flow physics of thick airfoils and build a foundation for validating numerical tools and informing the design and operation of next-generation, large-scale wind turbines.

## 2 Experimental Design

In their study, De Tavernier and von Terzi (2022) defined the boundary between the subsonic and transonic flow regimes for the FFA-W3-211 airfoil. This boundary was determined in terms of the inflow Mach number, $Ma_\infty$, and the angle of attack, AoA, using isentropic flow theory combined with the Prandtl–Glauert compressibility correction and XFoil simulations. Based on this transonic envelope, OpenFAST simulations revealed that the tip airfoil of the IEA 15 MW RWT, when operating near cut-out wind conditions and under high free-stream turbulence levels, can experience large negative AoAs ($-10°$ to $-15°$) at moderately high subsonic Mach numbers ($Ma_\infty \approx 0.3$). Under these conditions, the tip airfoil may intermittently enter the transonic flow regime.

Vitulano et al. (2025a) employed URANS simulations to test the predictions of the XFoil-based transonic envelope calculations. This is illustrated in Fig. 2, where the transonic envelope is represented by the solid black line. Here, the red circles highlight cases where local supersonic flow is detected, but no shock waves occur. The green squares denote cases where shock waves occur. It is also important to note that the URANS simulations were conducted for a fully turbulent boundary layer over the airfoil. Interestingly, it is evident from the same figure that the Reynolds number (Re) is also a crucial parameter for determining the onset of shock waves, apart from the inflow Mach number and angle of attack.

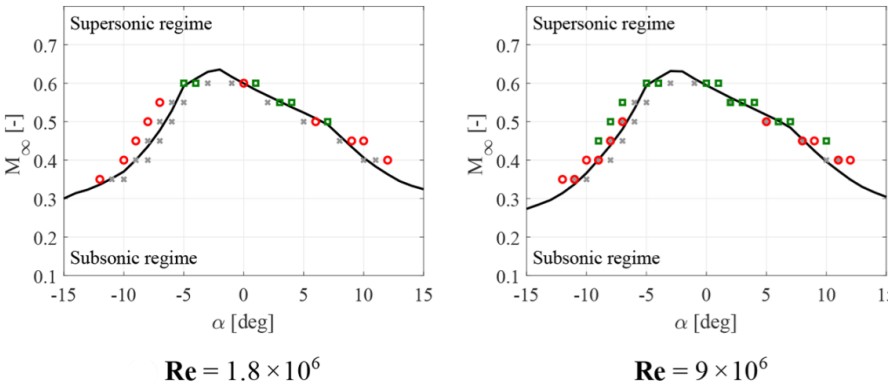

**Figure 2.** Subsonic-transonic boundary for the FFA-W3-211 airfoil generated using XFoil (solid black line), with symbols representing URANS simulations showing only subsonic flow (grey crosses), local supersonic regime established (red circles), and configurations in which shock waves appear (green squares) for different Reynolds numbers; from Vitulano et al. (2025a).

Notably, at a Re of $1.8 \times 10^6$ (close to that of the present experiments), shock waves only start appearing at an inflow Mach number of 0.6. However, when Re increases to $9 \times 10^6$ (similar to full-scale wind turbines), shock waves are detected already at
90  Ma $\approx 0.45$. Furthermore, hysteresis effects on pitching airfoils can further lower this threshold to Ma $\approx 0.35$ at Re $\sim \mathcal{O}(10^7)$ (Vitulano et al., 2025b). These inflow Mach numbers closely approach those observed on the tips of large turbines such as the IEA 22 MW RWT, and corroborate the growing importance of accounting for compressibility and shock waves on large rotors. There is a physical explanation for the Reynolds number influence on the occurrence of shock waves observed in the URANS simulations. For a turbulent boundary layer, a higher Re results in an increased resilience to separation under adverse pressure
gradients (Dróżdż et al., 2021), such as those imposed by high angles of attack. This means that at higher Re, the airfoil is better able to maintain its effective camber and promote high accelerations necessary for producing shock waves. Also, this effect is especially pronounced at steep inclinations, where the boundary layer is more prone to separation.

The Reynolds number dependency of the onset of shock waves has important implications for the experimental design. In our current wind tunnel test facility, a maximum Re $\sim 10^6$ can be achieved (see Section 3.1 for more details). It is seen from
Fig. 2 that a combination of Re $\sim 10^6$, Ma$_\infty \sim 0.3 - 0.4$, and any AoA cannot produce shock waves. It is, however, possible for shock waves to occur if Ma$_\infty$ is raised to 0.6 in the wind tunnel. Thus, due to the Re dependency, we require a higher inflow Mach number in our experiments to simulate equivalent transonic flow physics (i.e., occurrence of shock waves) that we would expect on a full-scale wind turbine blade at lower inflow Mach numbers but significantly elevated Reynolds numbers. Effectively, all three parameters, i.e., Re, Ma$_\infty$, and AoA, determine the onset of supersonic flow and shock waves in our case.
Consequently, we choose to perform our measurements at Ma$_\infty$ of 0.5 and 0.6 for Re $\sim \mathcal{O}(10^6)$. This allows us to confirm predictions made by Vitulano et al. (2025a) and shown in Fig. 2 regarding the transition to transonic flow with (Ma$_\infty = 0.6$) and without (Ma$_\infty = 0.5$) shock waves as the AoA is reduced for the same wind speed.

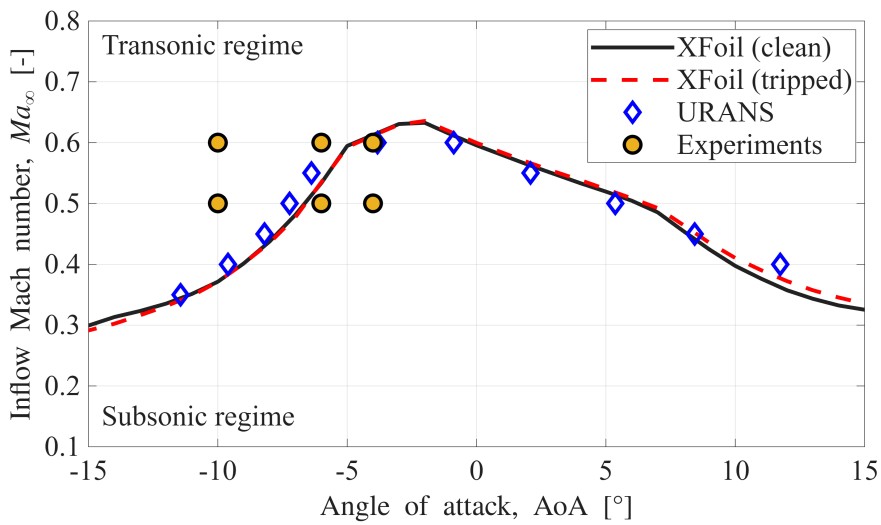

**Figure 3.** The transonic envelopes showing the separation between complete subsonic flow and transonic flow (i.e., with pockets of local supersonic flow) over the FFA-W3-211 airfoil as a function of the inflow Mach number ($\text{Ma}_\infty$) and the angle-of-attack (AoA), all at a Reynolds number of $1.8 \times 10^6$. Different envelopes pertain to different simulation methods: XFoil with transitioning boundary layer, i.e, clean airfoil (solid black line), XFoil with fully turbulent boundary layer, i.e., tripped airfoil (red dashed line), and URANS with fully turbulent boundary layer (blue diamonds). Yellow circles represent the experimental data points selected for the current study.

The choices of AoA are then decided based on the transonic envelope. For comparison, three different versions of the transonic envelope are presented in Fig. 3. All the transonic envelopes are for a Reynolds number of $1.8 \times 10^6$, i.e., close to the actual value of our experiments. Two different versions of the transonic envelope calculated with XFoil (and Prandtl-Glauert compressibility correction) are shown: one with a fully turbulent boundary layer (equivalent to a tripped airfoil with a fixed transition location, shown as a red dashed line) and the other with a freely-transitioning boundary layer (equivalent to a clean airfoil, shown as a solid black line). These two display no remarkable differences, except at high AoAs. The third transonic envelope is based on URANS simulations conducted at the same Reynolds number of $1.8 \times 10^6$, with a fully turbulent boundary layer, marked as blue diamonds. Compared to the XFoil calculations, the URANS envelope shows a more conservative prediction at negative AoAs steeper than $-6°$ and positive AoAs steeper than $8°$. However, for AoAs between $-5°$ to $+5°$, the URANS envelope predicts transonic flow to occur at lower inflow Mach numbers compared to the XFoil envelopes.

The experiments in the current study have been conducted on a clean airfoil model, given that there is no significant difference in the prediction of transonic flow occurrence for clean versus tripped airfoils (according to XFoil). Three geometric angles of attack (AoAs) for each inflow Mach number are tested: $-4°$, $-6°$, and $-10°$. The selected combinations of inflow Mach number and AoA allow investigation of the transition from subsonic to transonic flow, either through increasing negative AoA at a fixed Mach number (e.g., $-6°$ to $-10°$ at $\text{Ma}_\infty = 0.5$) or increasing Mach number at a fixed AoA (e.g., $\text{Ma}_\infty = 0.5$ to $0.6$ at $-6°$).

**Table 1.** Experimental conditions.

| Parameter | Value(s) | Unit |
|---|---|---|
| Inflow Mach number ($\mathrm{Ma}_\infty$) | 0.5 & 0.6 | - |
| Inflow Mach number (corrected) | 0.51 & 0.62 | - |
| Inflow velocity ($U_\infty$) | 166 & 197 | m/s |
| Chord-based Reynolds number (Re) | $1.4 \times 10^6$ & $1.6 \times 10^6$ | - |
| Total pressure ($p_0$) | 2.0 | bar |
| Total temperature ($T_0$) | 288 | K |
| Angle of Attack (AoA) | -4, -6 & -10 | $^\circ$ |
| Model chord | 67 | mm |
| Model span | 280 | mm |

The AoA and Mach number values that are reported in the subsequent discussion are those without corrections for wall interference and blockage effects, resulting from the finite test section height and model size. Blockage corrections according to Herriot (1947) suggest that the effective Mach number in the test section is up to 3% higher than nominal values (see Table 1); moreover, AoA corrections were not applied. While these uncertainties slightly shift the reported test conditions, the overall flow physics is not expected to deviate significantly from full-scale behavior. For clarity, all results are presented in terms of uncorrected $\mathrm{Ma}_\infty$ and geometric AoA.

## 3  Methodology

### 3.1  Wind Tunnel Facility

Measurements on a static FFA-W3-211 airfoil have been performed in the TST-27 transonic-supersonic blowdown-type wind tunnel at the Delft University of Technology (Fig. 4). In the transonic mode of operation, the free-stream Mach number in the test section is controlled by a choke mechanism downstream of it and can be varied in the range $0.5 \pm 0.01$ to $0.85 \pm 0.01$. The test section is 255 mm $\times$ 280 mm in height and width, respectively, with transparent windows for optical access present in both sidewalls. The total pressure in the tunnel can range from $1.5 - 4$ bar, which allows variation of the Reynolds number independent of the Mach number, and is set to $p_0 = 2$ bar in the current experiments. The total temperature, which is not actively controlled, is $T_0 = 288$ K. For the current study, Mach numbers of $0.5 \pm 0.01$ and $0.6 \pm 0.01$ are considered, which correspond to free-stream velocities of 166 and 197 m/s, respectively.

### 3.2  Wind Tunnel Model

The tip airfoil used in the IEA 15MW and 22MW RWTs is the FFA-W3-211 (see Fig. 1), belonging to the DTU FFA series (Bertagnolio et al., 2001). A model of the airfoil with a chord ($c$) of 67 mm was used for the tests, with a maximum thickness-

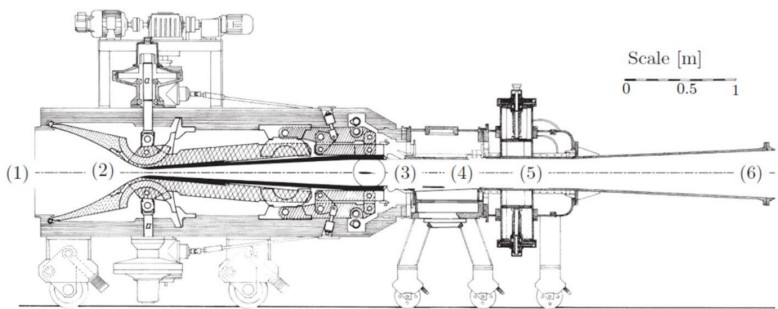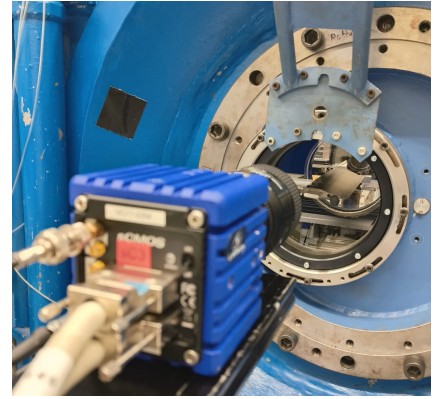

**Figure 4.** A schematic of the Transonic-Supersonic Wind Tunnel (TST-27) (left), showing (1) settling chamber, (3) test section, (5) choke, and (6) diffuser; a picture of the airfoil model installed in the test section of TST-27 (right).

to-chord ratio ($t_{max}/c$) of $21\%$. The model spans the entire width of the test section, resulting in an aspect ratio of more than 4, to approach 2D aerodynamic behavior. At the maximum inclination of $10°$, the geometric blockage ratio is $\approx 6\%$. The AoA of the model is adjusted manually using a digital angle gauge with an uncertainty of $\pm 0.1°$, and the values reported throughout refer to the geometric AoA with respect to the conventional orientation, which would be different from the actual AoA in the test section due to wall interference effects on the streamline curvature.

### 3.3 Schlieren Imaging

Schlieren imaging maps the gradient of the refractivity of a medium, which can be interpreted as a visualization of gradients of density and is thus useful for identifying compressible flow features such as shock and expansion waves. For the current study, schlieren is used as a preliminary analysis tool to obtain a quick and qualitative impression of the general flow-field, especially shock waves. A Z-type schlieren setup is employed, using a white LED with a 1 mm diameter pinhole for illumination, and images are acquired at a rate of 100 Hz using a LaVision Imager sCMOS at a cropped resolution of $1920 \times 1038$ pixels, corresponding to a field of view of 112 mm $\times$ 61 mm in the streamwise and vertical extent. The exposure time is maintained at 9 $\mu$s to avoid blurring of the shock motion.

### 3.4 Particle Image Velocimetry

Particle Image Velocimetry (PIV) is used in a planar configuration to measure two components of the velocity field at the spanwise center plane of the airfoil model. Given the full-field and quantitative measurement capabilities of PIV, it is employed as the primary diagnostic tool in this investigation to study the occurrence of transonic effects. For seeding the flow, DEHS (Di-Ethyl-Hexa-Sebacat) particles are used with an average diameter of 1 $\mu$m. For illuminating the particles, an Nd:YAG laser, with a wavelength of 532 nm, is shaped into a light sheet of approximately 1.5 mm thickness and projected along the spanwise center of the TST-27 test section, operating at a repetition rate of 15 Hz. Two LaVision Imager sCMOS cameras equipped with

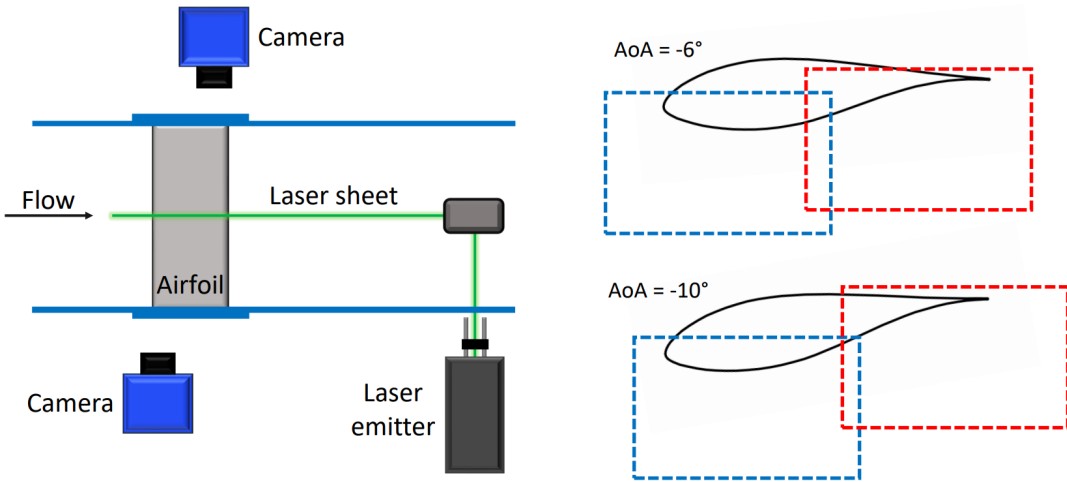

**Figure 5.** PIV setup details: cameras and laser arrangement around the test section (left) and approximate fields-of-view for two different inclinations (right).

Nikon Nikkor 105 mm lenses are used with an f-stop of 8 on either side of the test section to capture the particle images at an acquisition frequency of 15 Hz, with an overlap between the two fields-of-view that are combined to create a total field-of-view covering $\sim$ 98 mm $\times$ 55 mm in the streamwise and transverse directions, respectively, at a scale of approximately 49 px/mm. This translates to a total field of view spanning $146 \times 82$ % of the chord. In total, 1200 snapshots are recorded for each experimental configuration, to ensure convergence of flow statistics. The synchronization between the cameras and the laser is achieved using a LaVision Programmable Timing Unit (PTU) controlled by a PC using LaVision DaVis. The acquired raw images are then processed in LaVision DaVis to obtain the velocity fields, which have a velocity vector pitch of 0.16 mm, in both streamwise and vertical directions. A schematic of the experimental setup is presented in Fig. 5. Further post-processing of the velocity fields has been carried out using MATLAB.

### 3.5 Local Mach number calculations

Under the assumption of adiabatic flow (i.e. a constant value of the total temperature), the energy equation allows the local Mach number to be related to the local velocity magnitude, $U$, and the total temperature of the flow, $T_0$, according to:

$$\mathrm{Ma} = \frac{U}{\sqrt{\gamma R T_0 - \left(\frac{\gamma - 1}{2}\right) U^2}} \tag{1}$$

The total temperature is determined as the temperature that is measured in the settling chamber of the wind tunnel (see Fig. 4), using a thermocouple with an accuracy of $\pm 1°$C. During the experiments, the measured total temperature varied in the range 15-17°C. The overall results were not found to be remarkably sensitive to small variations of the total temperature in the given range; hence, an average value of $T_0 = 288$ K has been used throughout. Since both streamwise and vertical velocity

components, $(u, v)$, are obtained from the post-processed PIV measurements, it is relatively straightforward to determine the local Mach numbers using Equation 1, with $U = \sqrt{u^2 + v^2}$.

## 3.6 Uncertainty Quantification

The experimental measurements are affected by several uncertainties, and these are estimated below. The aim of this inventory is to show that the uncertainty values are sufficiently small compared to the mean values of the measurements. Thus, conclusions can be confidently derived from the trends seen in the mean and standard deviations of the measurements. The uncertainty estimates are tabulated in Table 2, and are further explained in the following text.

**Table 2.** Sources of uncertainty.

| Source | Value | Unit |
|---|---|---|
| Mean of total velocity, $\varepsilon_{\bar{U}}$ | $\leq 2.87$ | m/s |
| Standard deviation of total velocity, $\varepsilon_{\sigma_U}$ | $\leq 2.02$ | m/s |
| Mean of local Mach number, $\varepsilon_{\mathrm{Ma}}$ | $\leq 0.01$ | - |
| Standard deviation of local Mach number, $\varepsilon_{\sigma_{\mathrm{Ma}}}$ | $\leq 0.005$ | - |
| Cross-correlation, $\varepsilon_{cc}$ | $\leq 2.03$ | m/s |
| Spatial resolution, $\varepsilon_{sr}$ | $\leq 1\%$ | - |

Since the ensemble size used for calculating statistical quantities is finite, it leads to a statistical convergence uncertainty related to both the mean and standard deviation estimates. This is quantified using the standard deviation, $\sigma_u = \sqrt{u'^2}$, and ensemble size, $N$, (Benedict and Gould, 1996). The result for the mean value is:

$$\varepsilon_{\bar{u}} = \frac{\sigma_u}{\sqrt{N}} \tag{2}$$

And on the standard deviation itself:

$$\varepsilon_{\sigma_u} = \frac{\sigma_u}{\sqrt{2N}} \tag{3}$$

To estimate an upper limit to the statistical uncertainties, the maximum standard deviation value in the flow-field is used to calculate the same. In the current study, the acquisition rate (15 Hz) is sufficiently low to consider subsequent PIV snapshots as uncorrelated, which means that we can use the total number of snapshots, $N = 1200$, to calculate the statistical uncertainties.

Another uncertainty arises from the cross-correlation procedure employed to calculate velocities from the particle image pairs. For planar PIV, the uncertainty, $\varepsilon_{corr}$, is estimated to be 0.1 pixels. It can be further translated in terms of uncertainty in instantaneous velocity as (Humble, 2009):

$$\varepsilon_{cc} = \frac{\varepsilon_{corr}}{M \cdot \delta t} \tag{4}$$

where $M = 0.32$ is the magnification in the current setup and $\delta t$ is the laser pulse separation time.

Finally, the window size ($WS$) used for cross-correlation allows resolving flow structures up to a certain limit, which is represented by a wavelength $\lambda$. The resulting uncertainty is modelled using a sinc function, as shown by Schrijer and Scarano (2008).

$$\varepsilon_{sr} = \frac{u}{u_0} = \text{sinc}\left(\frac{WS}{\lambda}\right) \tag{5}$$

A multi-step correlation procedure, also employed in the current study, makes this uncertainty less pronounced. Also, given that the smallest resolvable flow structures are twice the window size (De Kat and Van Oudheusden, 2012), it is safe to approximate that $\epsilon_{sr} \leq 1\%$.

## 4  Results

In all the figures presented henceforth, the flow direction is from left to right.

### 4.1  Local Mach Number Trends

The first set of cases is shown in Fig. 6 for an inflow Mach number of $\text{Ma}_\infty = 0.5$. The two shallower angles of attack (AoA), $-4°$ and $-6°$, are predicted to remain fully subsonic according to the transonic envelope in Fig. 3. This is confirmed by the corresponding mean local Mach number distributions in Fig. 6. For $\text{AoA} = -4°$ (Fig. 6a), the maximum local Mach number reaches approximately $0.8$ near the airfoil surface at the point of maximum thickness. A similar trend is observed for $\text{AoA} = -6°$ (Fig. 6c), though a larger portion of the flow accelerates to this maximum value. Additionally, the flow accelerates more rapidly near the leading edge due to a stronger suction peak associated with the increased incidence.

At the steepest angle of attack, $\text{AoA} = -10°$, the flow near the leading edge accelerates to nearly sonic conditions in the mean flow (Fig. 6e). According to the transonic envelope (Fig. 3), this configuration lies well within the transonic regime. However, the mean Mach number field suggests only a tiny region of potential supersonic flow at the leading edge.

None of these cases achieves sustained sonic conditions, as can be inferred from the standard deviation of the local Mach number. Even for $\text{AoA} = -10°$, the maximum standard deviation near the leading edge is only about $0.05$ (Fig. 6f). Combined with a mean local Mach number near unity, this indicates that brief, localized supersonic pockets may intermittently form at $\text{Ma}_\infty = 0.5$, $\text{AoA} = -10°$. This possibility is examined further in Section 4.2. For the other two cases, there is no evidence of supersonic flow in either the mean or standard deviation fields, consistent with the transonic envelope predictions.

As the inclination increases, a strongly unsteady shear layer is seen to emerge in the flow. This is reflected in the standard deviation of the local Mach number for different inclinations. For $\text{AoA} = -6°$, the fluctuating shear layer appears to start from $x/c \approx 0.5$ (Fig. 6d); for the steeper $\text{AoA} = -10°$, it starts more upstream at $x/c \approx 0.4$ (Fig. 6f).

Steep angles of attack generate strong adverse pressure gradients, leading to significant flow separation over the airfoil. For $\text{AoA} = -10°$, a large separated region is visible in the mean flow-field, as highlighted by the white dashed line in Fig. 6e. The white dashed line represents the contour of zero streamwise velocity, which is used as a proxy to indicate separated flow regions in this study. It is to be noted that this provides only a conservative estimate of the recirculation region. In contrast,

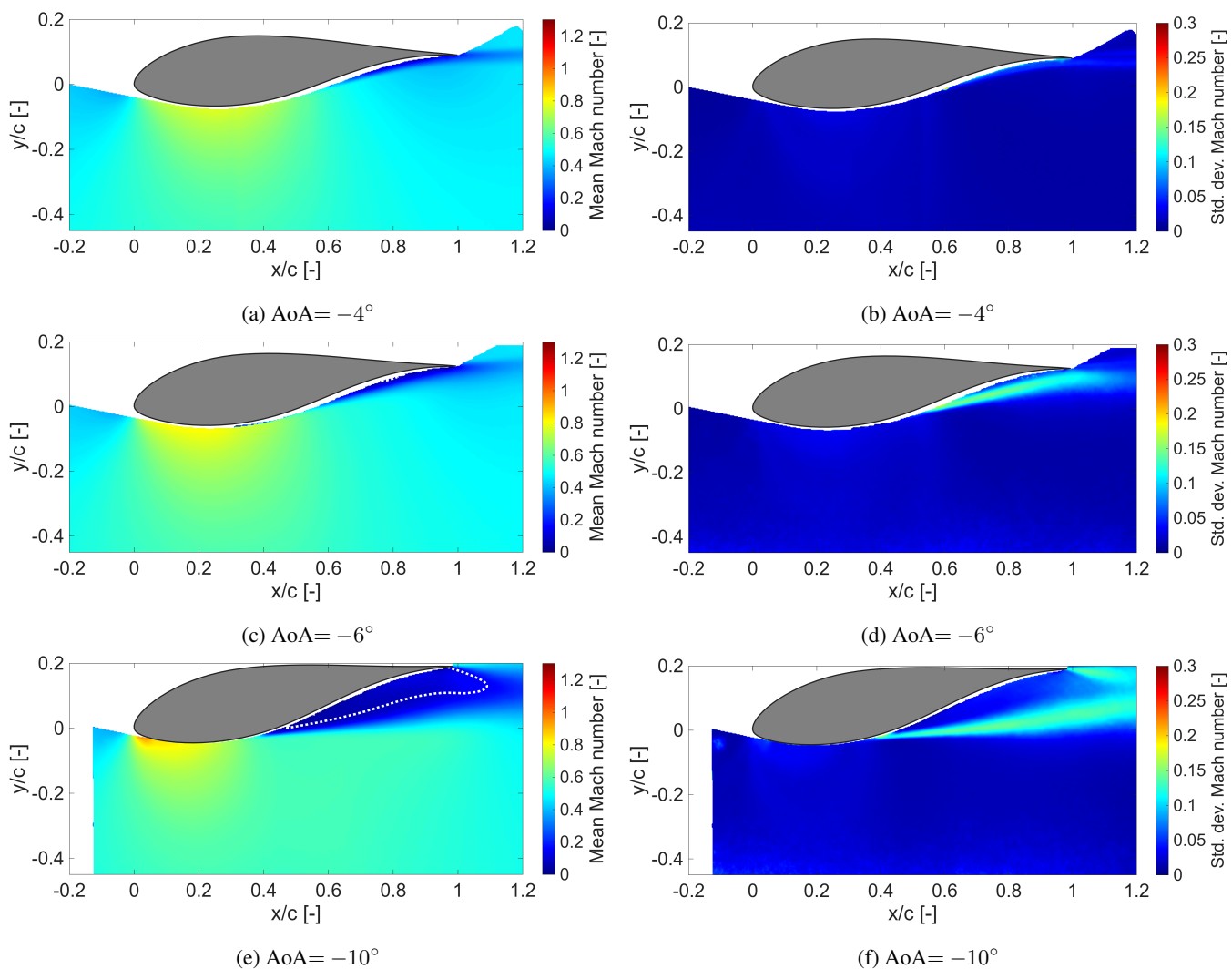

**Figure 6.** Contours of Mach number: mean values (left column) and standard deviation (right column) for $\mathrm{Ma}_\infty = 0.5$. The white dotted line shows zero streamwise (x) velocity.

the shallower angles of attack show no such separation. This separation reduce the effective camber, a factor that will play an important role in the next set of results.

In Fig. 7, the mean and standard deviation of the local Mach number are presented for the higher free-stream Mach number of
0.6, for the same AoA values. A mean local Mach number equal to unity is marked with a solid black line, while a white dashed line indicates zero streamwise velocity. Note that the contour scales are consistent across Figs. 6 and 7 to assist comparison.

Now, the free-stream is energetic enough to result in local supersonic flow regions (transonic flow) owing to the suction peak generated at the airfoil leading edge. All three inclinations show supersonic flow pockets, increasing in extent with

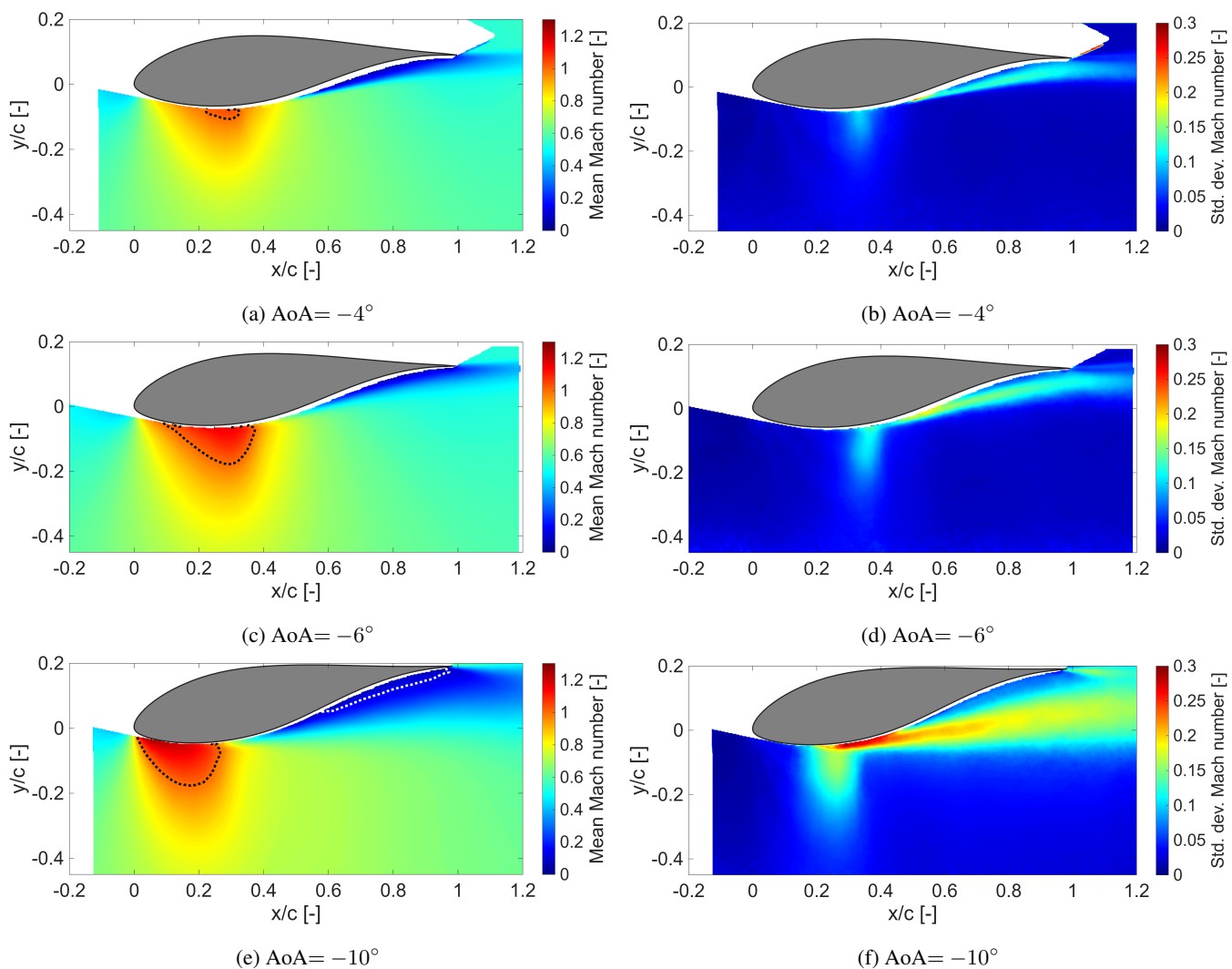

**Figure 7.** Contours of Mach number: mean values (left column) and standard deviation (right column) for $Ma_\infty = 0.6$. The dotted black line represents a local Mach number of 1. The dotted white line indicates zero streamwise (x) velocity.

increasing steepness. From basic transonic flow physics, it is expected that upon increasing the angle-of-attack, the shock that terminates the supersonic region shifts more downstream (Tijdeman and Seebass, 1980). However, going from $AoA = -6°$ to $AoA = -10°$ a contrary trend is revealed, i.e., the supersonic flow pocket (in the mean flow) terminates more upstream even when the inclination is steeper.

This 'inversion' of the supersonic region movement towards the leading edge upon increasing the AoA beyond a critical value is a necessary condition for the onset of transonic buffet, as described by (Pearcey, 1958). It is a consequence of the trailing-edge separation in the mean flow due to a high inclination of $-10°$. As discussed previously, such a steep inclination

already results in large-scale flow separation even without local supersonic flow, as was observed for the lower $Ma_\infty = 0.5$ at $AoA = -10°$ in Fig. 6e. This causes a decrease in the effective camber of the airfoil. Upon increasing the inflow Mach number to 0.6, local supersonic flow is produced, which terminates prematurely to match the pressure imposed by the incipient trailing-edge separation at $-10°$. This highlights the uniqueness of the current conditions, notably, the steep inclination and the resulting trailing-edge separation. Such extreme camber/inclinations and the resulting incipient trailing-edge separation are not encountered in typical transonic flow studies for supercritical airfoils (D'Aguanno et al., 2021; Accorinti et al., 2022). In the present case, trailing-edge separation is already present at non-transonic conditions, and the supersonic flow pocket has to adjust to it. This also raises interesting questions on the resulting shock-separation interaction under such conditions, explored further in Section 4.4.

Another interesting point of comparison between $Ma_\infty = 0.5$ and $Ma_\infty = 0.6$ at $AoA = -10°$ is the decrease in mean-flow separation extent upon increasing $Ma_\infty$. This is evident from observing the white dotted lines enclosing the region of separated flow in Figs. 6e and 7e. The reason for this is that the oscillations of the shock wave in the case of $Ma_\infty = 0.6$ interact with the separated flow, causing large variations in instantaneous flow separation similar to that observed in typical transonic buffet cycles (D'Aguanno et al., 2021). When the shock is more downstream compared to its mean location, flow separation is significantly reduced. Hence, there is also a reduction in separation in the mean flow. This is illustrated again in Section 4.2.

The standard deviation of the local Mach number shows that shear-layer fluctuations intensify with increasing AoA. Comparing Figures 6 and 7 also reveals that higher inflow Mach numbers amplify these fluctuations at the same AoA. At $Ma_\infty = 0.6$, all cases exhibit a region of relatively high standard deviation ($\sigma_{Ma} \approx 0.1-0.15$) extending transversely near the maximum thickness, as shown in Figures 7b, 7d, and 7f. This region also aligns with the corresponding downstream edge of the local supersonic zone in the mean flow.

For example, for $AoA = -10°$ (Fig. 7f), $\sigma_{Ma}$ reaches $\sim 0.15$ between $x/c \approx 0.15-0.35$ and $y/c \approx -0.2$. Within the same area, the mean Mach number ranges from 0.9 to 1.2 (Fig. 7e), implying instantaneous values between $\sim 0.75$ and $\sim 1.35$. This suggests that instantaneously, the flow at the location might be either subsonic or supersonic, possibly due to being traversed by an unsteady shock wave. Instantaneous Mach number contours are used in Section 4.3 to further investigate this behaviour.

Similar trends appear for $AoA = -6°$ and $-4°$, with intermittent transitions between subsonic and locally supersonic states suggested by the mean and standard deviation fields. Notably, at $-4°$, the transonic envelope (Fig. 3) predicts no transonic flow at $Ma_\infty = 0.6$, yet both the mean and standard deviation fields indicate its presence and unsteady shock wave (Figures 7a, 7b). This discrepancy highlights uncertainties in the envelope predictions, suggesting either a lower actual envelope or an increased effective inflow Mach number due to blockage, or both.

## 4.2 Probability of Supersonic & Separated Flow

While the mean and standard deviation of the Mach number proved insightful in understanding the features of transonic flow over the FFA-W3-211 airfoil, they are not sufficient to characterize the unsteadiness associated with shock waves and flow separation. To investigate flow intermittency in terms of supersonic and separated flow, their probability of occurrence in the flow-field is calculated. This represents how often a point in the flow-field experiences supersonic/separated flow, and is simply

the ratio of the number of snapshots with supersonic/separated flow at a given location, relative to the total number of snapshots:

$$P_{\text{Ma}>1}(x,y) = \frac{n_{\text{Ma}>1}(x,y)}{N}, \tag{6}$$

$$P_{u<0}(x,y) = \frac{n_{u<0}(x,y)}{N}, \tag{7}$$

where $P_{\text{Ma}>1}$ and $P_{u<0}$ represent the probability of supersonic and separated flow, respectively, $n$ denotes the number of snapshots exhibiting supersonic or separated flow at a given location $(x,y)$, and $N$ is the total number of snapshots. Thus, the

probability map also reveals the spatial extent of the investigated feature. The resulting probability fields for selected cases are shown in Fig. 8. Since at $\text{Ma}_\infty = 0.5$, AoAs of $-4°$ and $-6°$ did not exhibit any local supersonic flow, they are excluded from the present analysis. The red colormap is used to represent the probabilities of local supersonic flow, while the blue colormap shows that of separated flow. A 5% probability for each is denoted by the black dashed line, which is used as a reliable threshold to mark the full extent of supersonic/separated flow regions.

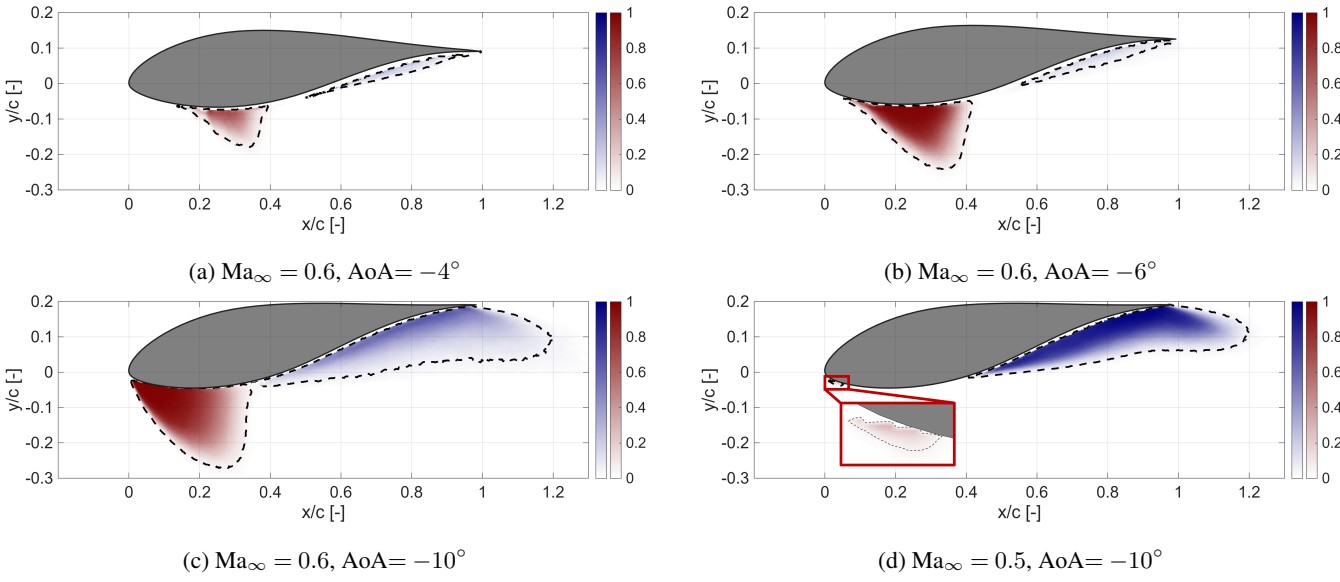

**Figure 8.** Distribution of the probabilities of supersonic flow ($P_{\text{Ma}>1}$, red colorscale) and separated flow ($P_{u<0}$, blue colorscale) for different configurations. A probability of 5% is marked with dashed black lines.

Starting with the probability of supersonic flow for different conditions, we see a steady growth in the extent of supersonic flow as the inclination is increased from $-4°$ (Fig. 8a) to $-6°$ (Fig. 8b) at $\text{Ma}_\infty = 0.6$, extending more upstream, downstream, and transversely. Also, at $-6°$, supersonic flow is seen to occur close to 100% in a significant region between $x/c \sim 0.1-0.25$. When the inclination is further steepened to $-10°$ at $\text{Ma}_\infty = 0.6$ (Fig. 8c), we see the region with supersonic flow shift more upstream compared to $-6°$. This was already remarked upon in the mean flow-fields for the same two cases, and the reason

for this is the large flow separation induced by the steepest inclination.

At this point, it is relevant to look at the probability of separated flow. For the lower inclinations of $-4°$ and $-6°$ at $\text{Ma}_\infty = 0.6$, there is no separation detected in the mean flow; however, the probabilities show that some intermittent separation occurs. However, instantaneous separation for these two cases appears to be only very minor and does not interact at all with the corresponding local supersonic flow pocket.

At the steepest inclination of $-10°$, instantaneous flow separation is seen to occur over a significant spatial extent. At $\text{Ma}_\infty = 0.5$, the 5% probability line of flow separation starts near $x/c \sim 0.4$ (Fig. 8d). In comparison, the 5% probability of separation starts from $x/c \sim 0.35$ for the elevated $\text{Ma}_\infty = 0.6$, as shown in Fig. 8c. The separated flow is also in close proximity to the supersonic flow region. This suggests that the unsteady shock wave might be interacting with the separated flow, pulling it more upstream intermittently.

Another clue supporting an unsteady shock-separation interaction is the difference in separated flow probability distributions between $\text{Ma}_\infty = 0.5$ and $0.6$ at $\text{AoA} = -10°$. It is clear that at the lower $\text{Ma}_\infty$, when no shock waves occur, a significantly larger region of the flow remains separated close to 100% of the time (Fig. 8d). When $\text{Ma}_\infty$ is increased, the probability of separation in the same spatial region drops (Fig. 8c), which indicates that an unsteady shock wave might be intermittently decreasing the extent of flow separation in sync with its downstream motion. As discussed earlier, this is a typical feature 310   observed in transonic buffet cycles (D'Aguanno et al., 2021).

    It was earlier noted that increasing $\text{Ma}_\infty$ from 0.5 to 0.6 at $\text{AoA} = -10°$ resulted in a decrease in mean flow separation. Here, the probabilities show that this does not translate to a decrease in flow separation in an instantaneous sense. Rather, the decrease is an artefact of the unsteady shock-separation interaction, which may instantaneously decrease or even increase the extent of flow separation compared to the no-shock case.

Finally, for $\text{Ma}_\infty = 0.5$ a tiny region of intermittent local supersonic flow is observed at $\text{AoA} = -10°$ in Fig. 8d. It is highlighted in the zoomed-in view at the leading edge, showing a $\sim 30\%$ probability of developing supersonic flow. However, it does not lead to any supersonic pockets in the mean flow, as seen in Fig. 6e.

    For comparative purposes, a cumulative probability metric to characterize the occurrence of supersonic flow is defined, which gives a single measure that combines the spatial extent and local probability strength. This metric is calculated by integrating 320   the probability value over the domain, as follows:

$$\overline{A_{\text{Ma}>1}} = \iint P_{\text{Ma}>1}(x,y) \cdot d\left(\frac{x}{c}\right) d\left(\frac{y}{t_{max}}\right). \tag{8}$$

    Similarly, a probability-weighted area of separated flow is also defined, using negative streamwise velocities as a proxy to approximate the separated flow region:

$$\overline{A_{u<0}} = \iint P_{u<0}(x,y) \cdot d\left(\frac{x}{c}\right) d\left(\frac{y}{t_{max}}\right). \tag{9}$$

The corresponding values of this probability-weighted area of supersonic flow ($\overline{A_{\text{Ma}>1}}$) and separated flow ($\overline{A_{u>0}}$) represent, respectively, the extent and frequency of supersonic and separated flow.

    Mathematically, the quantities defined in Equations 8 and 9 are equivalent to the mean of the instantaneous areas of supersonic and separated flow, respectively. The instantaneous areas of supersonic ($A_{sup}$) and separated flow ($A_{sep}$) are defined for

each snapshot as:

$$A_{sup} = \iint f_{\mathbf{Ma}>1}(x,y) \cdot d\left(\frac{x}{c}\right) d\left(\frac{y}{t_{max}}\right), \tag{10}$$

$$A_{sep} = \iint f_{u<0}(x,y) \cdot d\left(\frac{x}{c}\right) d\left(\frac{y}{t_{max}}\right), \tag{11}$$

where,

$$f_{\mathbf{Ma}>1} = \begin{cases} 1, & \text{if } \mathrm{Ma} > 1 \\ 0, & \text{if } \mathrm{Ma} \le 1 \end{cases}, \quad f_{u<0} = \begin{cases} 1, & \text{if } u \le 0 \\ 0, & \text{if } u > 0 \end{cases} \tag{12}$$

In Equations 8-11, Ma denotes the local Mach number while $u$ represents the streamwise velocity component. All quantities are, by definition, normalised by chord × max. thickness of the airfoil, $c \cdot t_{max}$. The corresponding values are tabulated in Table 3, for the configurations considered, providing a quantitative confirmation of the previous observations. The instantaneous separated flow area ($A_{sep}$) is encountered again when studying the shock-separation interaction in Section 4.4.

**Table 3.** Ssupersonic and separated flow region areas [% of chord × max. thickness, $c \cdot t_{max}$].

| Case | $\overline{A_{\mathbf{Ma}>1}}$ (= Mean $A_{sup}$) | $\overline{A_{u<0}}$ (= Mean $A_{sep}$) |
|---|---|---|
| $\mathrm{Ma}_\infty = 0.6$, AoA$= -4°$ | 2.14 | 0.47 |
| $\mathrm{Ma}_\infty = 0.6$, AoA$= -6°$ | 10.09 | 0.62 |
| $\mathrm{Ma}_\infty = 0.6$, AoA$= -10°$ | 13.62 | 9.95 |
| $\mathrm{Ma}_\infty = 0.5$, AoA$= -10°$ | 0.05 | 16.43 |

### 4.3 Occurrence of Shock Waves

So far, the mean flow-field, along with the standard deviations and probability of local supersonic flow, has hinted at the occurrence of unsteady shock waves in some cases. A quick and qualitative visualization of shock waves is obtained through schlieren imaging. In Fig. 9, we use instantaneous schlieren frames to compare two cases to see the distinction in flow-fields when shocks appear versus when they do not.

With the current schlieren setup, regions darker than the background gray show regions experiencing compressibility, such as compression waves and shock waves. Consequently, shocks can be clearly visualised for $\mathrm{Ma}_\infty = 0.6, \mathrm{AoA} = -10°$ (Fig. 9b) between $x/c = 0.2 - 0.4$. However, for $\mathrm{Ma}_\infty = 0.5, \mathrm{AoA} = -10°$ (Fig. 9a), no shocks are observed. These observations are in line with expectations based on the previous analysis of PIV results. It is also important to note that there appear to be multiple shock waves in Fig. 9b. The underlying reason is that schlieren produces a spanwise-integrated visualization, and a curved shock front appears as multiple shocks in the image. Schlieren images of additional cases can be found in Appendix B.

With PIV, the shock front is identified more unambiguously since the measurements correspond to a single plane, which is at the spanwise center of the model in this case. All cases discussed henceforth pertain to $\mathrm{Ma}_\infty = 0.6$, since the earlier

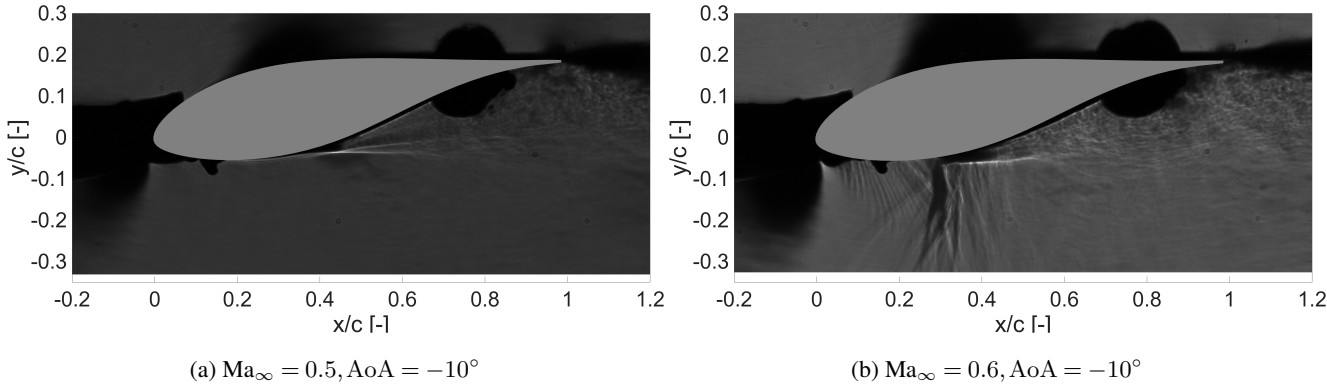

(a) $\mathrm{Ma}_\infty = 0.5, \mathrm{AoA} = -10°$  (b) $\mathrm{Ma}_\infty = 0.6, \mathrm{AoA} = -10°$

**Figure 9.** Instantaneous schlieren images showing no shock waves in (a) while clear shock waves appear in (b) between $x/c = 0.2 - 0.4$.

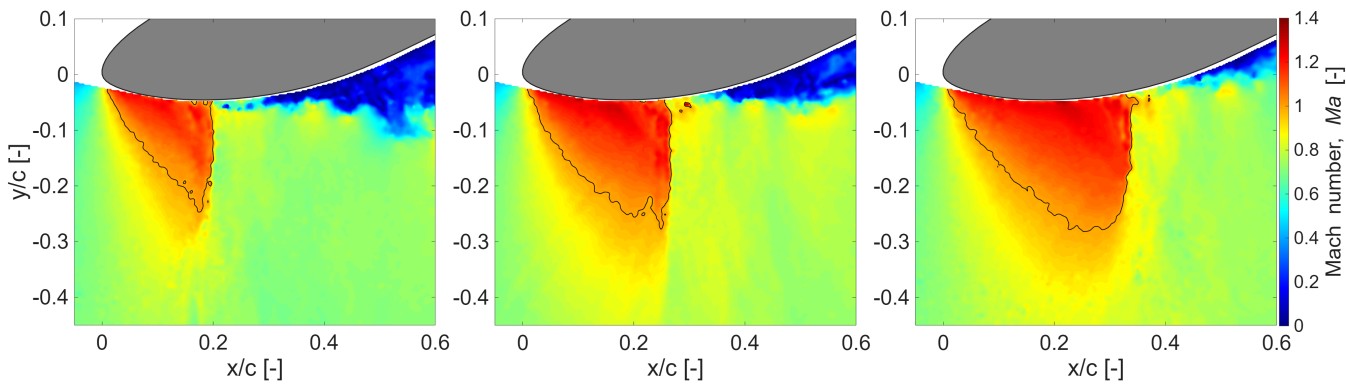

**Figure 10.** Instantaneous PIV frames for $\mathrm{Ma}_\infty = 0.6$ and AoA$= -10°$. A local Mach number of 1.0 is marked with a solid black line.

discussion already revealed that no shock waves were observed at $\mathrm{Ma}_\infty = 0.5$. At an AoA of $-10°$ and $\mathrm{Ma}_\infty = 0.6$, the shock wave also demonstrates a strongly unsteady nature, as evident from selected instantaneous Mach number contours shown in Fig. 10. The shock position is observed to vary between $x/c = 0.2$ in the leftmost frame to $x/c = 0.35$ in the rightmost frame. This is in good agreement with the region of high standard deviation in the local Mach number ($x/c = 0.15 - 0.35$) in Fig. 7f.

Simultaneously, the snapshots in Fig. 10 give further evidence of a high unsteadiness of the separated flow region, which appears to be related to the shock motion, as commonly observed in transonic buffet studies (D'Aguanno et al., 2021).

A shock wave detection procedure has been applied to the PIV snapshots (see Appendix A for details), to track the unsteady shock wave position frame-by-frame. At this low acquisition rate of the PIV data (15 Hz), subsequent PIV frames represent a random sampling of the shock motion cycle, i.e., two consecutive frames can have the shock being in completely different

phases of the oscillation cycle. Although this still allows a statistical characterization of the unsteady flow, the low acquisition rate implies that dynamical features such as the shock motion frequency, which is of a higher order, cannot be established. Instead, a probability density function (pdf) of the shock location for different cases is calculated to provide further insight into

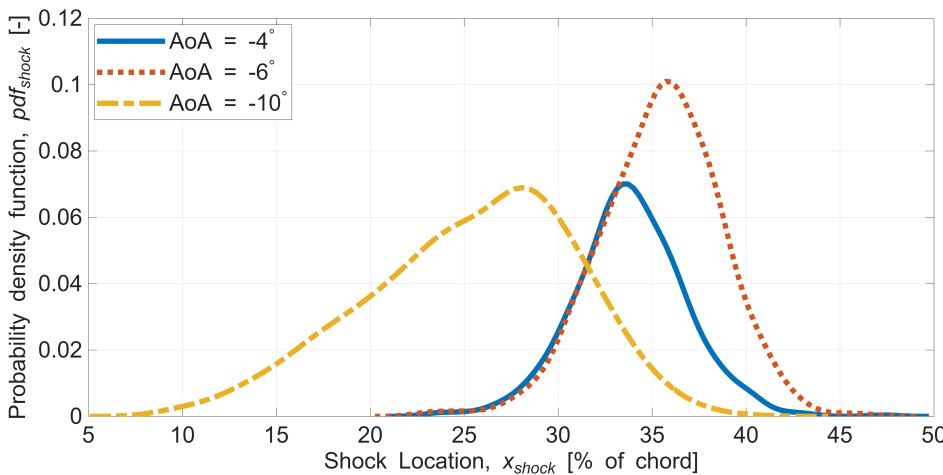

**Figure 11.** Probability distribution of shock locations for various angles of attack at $\mathrm{Ma}_\infty = 0.6$.

the shock dynamics. This is shown in Fig. 11. Here, it is worth mentioning that the pdf is normalized with respect to the total number of frames recorded in each case, and not by the total number of frames that exhibit a shock wave. Thus, the area under each curve is representative of the fraction of time that a shock wave is detected, for the particular configuration.

With the shallowest AoA of $-4°$, the shock pdf (solid blue line) is centered around 34% of the chord, with a range of $\approx 20\%$ of the chord. In comparison, when the AoA is slightly steeper to $-6°$ (red dotted line), the shock pdf peak shifts to $x/c \sim 36\%$ and a higher oscillation range $\approx 25\%$ of the chord. Thus, the mean shock location shifts downstream, and the oscillation range extends. At the steepest AoA of $-10°$, the shock pdf has a much flatter peak and is centered more upstream at $x/c \sim 27\%$, with a much wider spread: $x/c \sim 7 - 40\%$. The upstream shift at the steepest AoA was already expected based on the previous analysis of the mean flow-fields, owing to the large flow separation. The mean and standard deviation values of the shock location for the three cases are tabulated in Table 4.

Another noteworthy outcome from the shock wave detection is how frequently a shock wave is detected for the different cases. As discussed before, this can be calculated by integrating the areas under the pdfs to obtain the overall probability of shock occurrence as follows:

$$P_{shock} = \int pdf_{shock}(x) \cdot dx \tag{13}$$

In the same fashion, the overall probability of supersonic flow ($P_{sup}$) can be calculated as the ratio of the number of snapshots that contain a minimum threshold of supersonic vectors to the total number of snapshots recorded. Note that this is a different measure compared to $\overline{A_{\mathrm{Ma}>1}}$ and $A_{sup}$ presented in section 4.2 and Table 3, which were used to quantify the spatial extent of supersonic flow. In contrast, $P_{\mathrm{Ma}>1}$ simply reflects how often local supersonic flow is observed for a certain configuration, without taking into account the strength and size of this supersonic region.

It is interesting to compare the values of $P_{shock}$ and $P_{sup}$, as given in Table 4. For the shallowest AoA of $-4°$, local supersonic flow occurs 97% of the time; however, shock waves are detected only in 48% of the frames. Thus, the supersonic

**Table 4.** Characteristic properties of shock occurrence at $Ma_\infty = 0.6$.

| Case | Mean $x_{shock}$ [% of chord] | Std. dev. $x_{shock}$ [% of chord] | $P_{shock}$ [%] | $P_{sup}$ [%] |
|---|---|---|---|---|
| AoA$= -4°$ | 33.8 | 3.0 | 48 | 97 |
| AoA$= -6°$ | 35.3 | 3.2 | 79 | 100 |
| AoA$= -10°$ | 25.3 | 5.5 | 95 | 100 |

flow pockets do not converge into shock waves each time. The intermittency in shock occurrence is further illustrated in Fig. 12, where three instantaneous frames for $Ma_\infty = 0.6$ and AoA$= -4°$ are shown, and the local supersonic pocket is marked with a solid black line. The left and middle frames are seen to contain supersonic flow pockets, but these are small and do not terminate abruptly with a shock. However, the right frame contains a relatively larger local supersonic region, which culminates in a shock wave, as suggested by the nearly vertical downstream edge of the supersonic pocket and the abrupt drop in local Mach number.

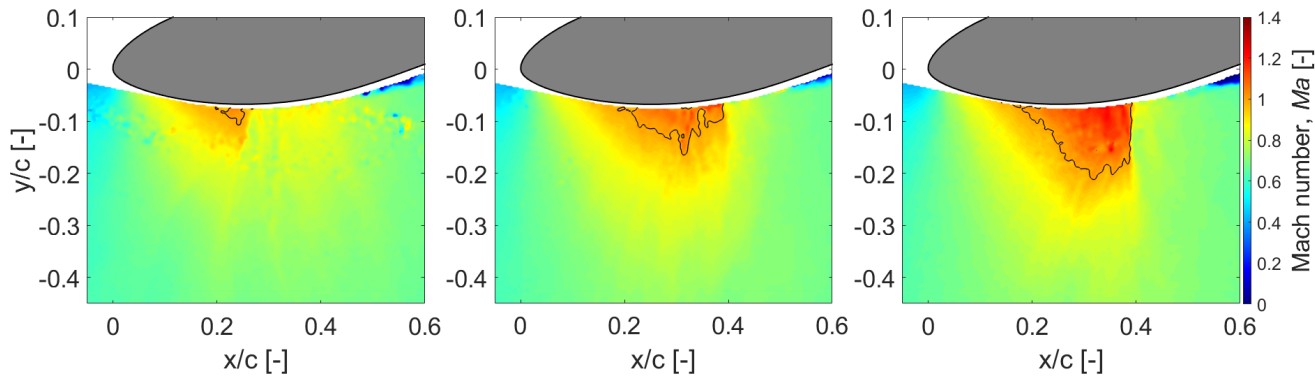

**Figure 12.** Instantaneous PIV frames for $Ma_\infty = 0.6$ and AoA$= -4°$. A local Mach number of 1.0 is marked with a solid black line. All three frames show a local supersonic flow region, but only the rightmost frame contains a shock (at $x/c \approx 0.4$).

For the two steeper AoAs, the probability of local supersonic flow is 100%, as noted in Table 4. However, for AoA$= -6°$, shock waves are observed around 79% of the time, whereas for the steepest inclination of $-10°$, 95% of the frames exhibit shock waves. In conclusion, the occurrence of supersonic flow forms no guarantee of a shock wave forming, especially for shallower AoAs, where shock wave occurrence could be intermittent rather than continuous.

### 4.4 Shock-separation interaction

As discussed in the previous sections, the mean flow-fields along with the probability of supersonic/separated flow already hint at the existence of unsteady shock waves. This was further confirmed in section 4.3. However, unsteady shock waves do not always lead to the establishment of shock-separation interaction in the current investigation. For example, there was no hint of instantaneous separation bubbles triggered by the shock waves suggested by the combined probability of supersonic/separated flow occurrence for $Ma_\infty = 0.6$, $AoA = -4°$ and $-6°$ (Fig. 8). The only case where shock-separation interaction seems a

possibility is $Ma_\infty = 0.6, AoA = -10°$. The primary reason for this is that the inclination is already steep enough to trigger trailing-edge separation without shock waves. However, it is unclear whether the shock waves are strong enough to induce separation by themselves, and any interaction is a result of the unsteady nature of the shock and its overlap with the already separated flow. In this section, the focus is on $Ma_\infty = 0.6, AoA = -10°$ to establish shock-separation interaction and to analyse the resulting unsteady flow-field.

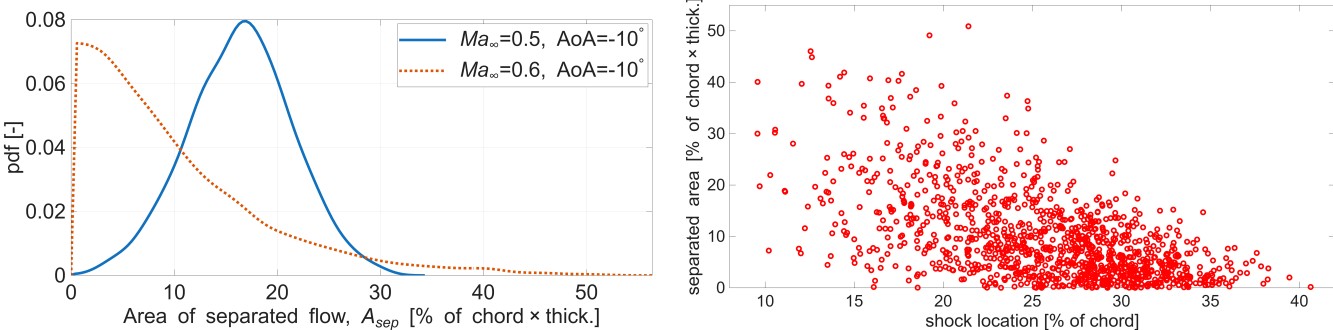

**Figure 13.** Probability distribution of instantaneous separated areas for $Ma_\infty = 0.5$ & $0.6$ at $AoA = -10°$.

**Figure 14.** Shock location versus separation area for $Ma_\infty = 0.6, AoA = -10°$.

First, the probability distribution of the instantaneous separated flow area ($A_{sep}$, as defined in Equation 11) is considered for $Ma_\infty = 0.5$, $AoA = -10°$, a case without mean supersonic flow and hence without shock waves. The corresponding distribution is shown in Fig. 13 (blue line). The distribution exhibits a Gaussian nature with the mean occurring at $3.52\%$ of $c \cdot t_{max}$. In the same figure, the distribution of the instantaneous separation area for $Ma_\infty = 0.6$, $AoA = -10°$ is also presented. At this higher Mach number, the probability distribution of the separated area is markedly different. This behavior is attributed to the occurrence of shock–separation interaction, since shock waves arise at $Ma_\infty = 0.6$. With the presence of shocks, the separated area is observed to be lower in most instances compared to the subsonic case.

The existence of shock–separation interaction for $Ma_\infty = 0.6$, $AoA = -10°$ is further established by plotting the instantaneous shock location against the separated area, as shown in Fig. 14. When the shock is located further downstream, the separated area is typically low. In contrast, when the shock is positioned further upstream, the separated area spans a broader range, from low to high values. This confirms that the shock location strongly influences the extent of flow separation in this case, with a trend that is in good qualitative agreement with the visualizations in Fig. 10. However, a similar correlation between shock position and separation area size is not observed for other inclinations ($-4°$ and $-6°$) at $Ma_\infty = 0.6$, despite the occurrence of shocks. A straightforward explanation is that in these cases, neither the shock strength nor the inclination is sufficient to trigger large-scale separation over the airfoil. The corresponding plots are provided in Appendix C.

Next, a phase-averaging procedure is employed to establish a more complete picture of the shock-separation interaction observed for $Ma_\infty = 0.6, AoA = -10°$. A crucial characteristic of the interaction is obtained from high-speed schlieren imaging, acquired at 7.2 kHz. Consecutive frames representing a time-resolved shock movement reveal that the extent of flow separation decreases when the shock is in the phase of its downstream motion, as shown in the left column of Fig. 15. On the other hand,

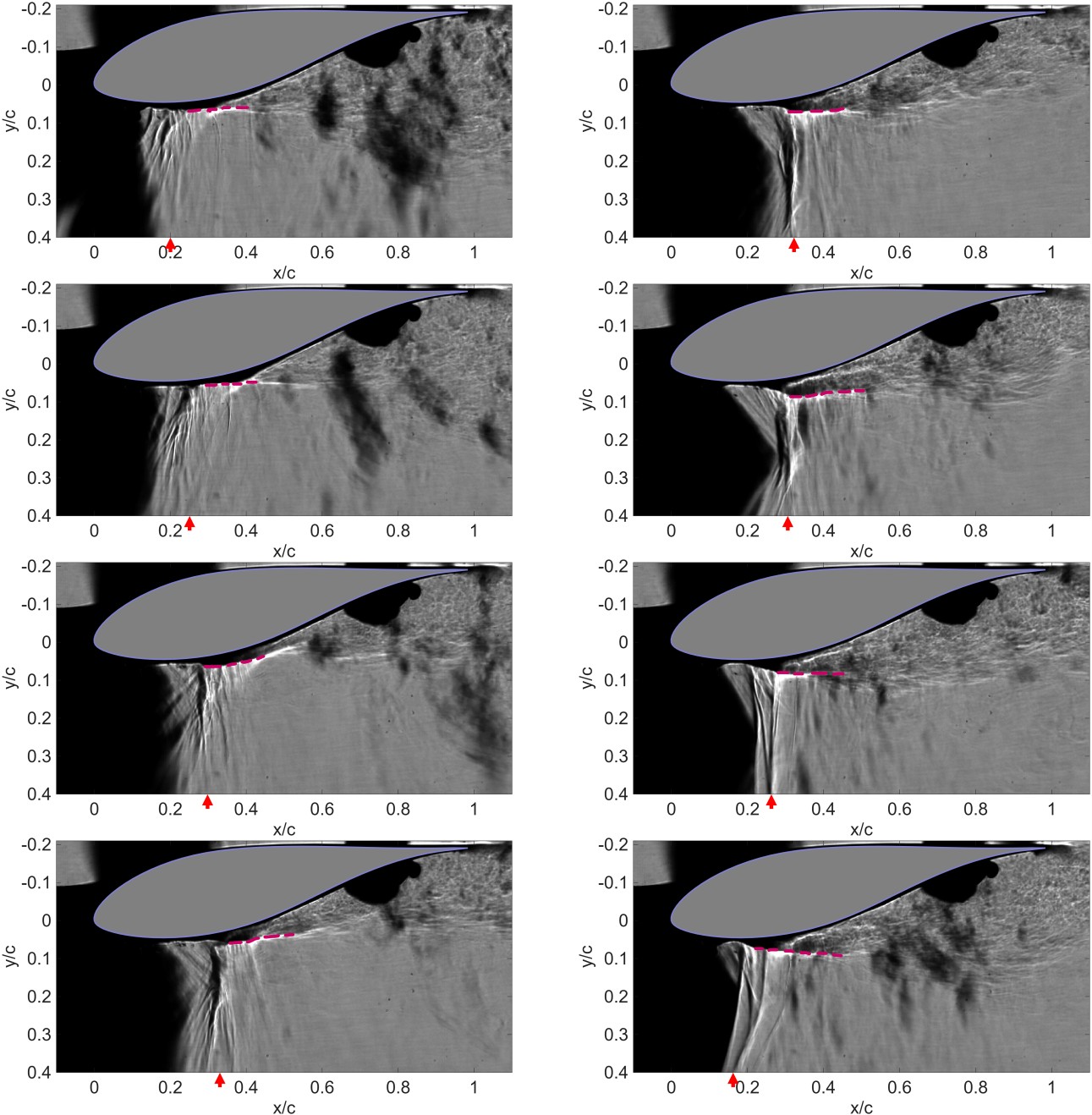

**Figure 15.** High-speed schlieren images acquired at 7.2 kHz for $\mathrm{Ma}_\infty = 0.6, \mathrm{AoA} = -10°$, showing the shock traversing downstream (from top to bottom). The estimated edge of the shear layer is marked with a dashed purple line, and the approximate shock edge is shown with a red arrow at the abscissa.

**Figure 16.** High-speed schlieren images acquired at 7.2 kHz for $\mathrm{Ma}_\infty = 0.6, \mathrm{AoA} = -10°$, showing the shock traversing upstream (from top to bottom). The estimated edge of the shear layer is marked with a dashed purple line, and the approximate shock edge is shown with a red arrow at the abscissa.

separation increases when the shock moves upstream, shown in Fig. 16. This asymmetry in the transonic buffet behavior was already reported earlier and has been studied in detail by D'Aguanno et al. (2021) in the transonic buffet cycle for a supercritical airfoil. This characteristic allows for resolving the shock motion direction ambiguity even in non-time-resolved measurements. Thus, the upstream and downstream movement of the shock can be differentiated in the low-speed (15 Hz) PIV data, based on the extent of the separation.

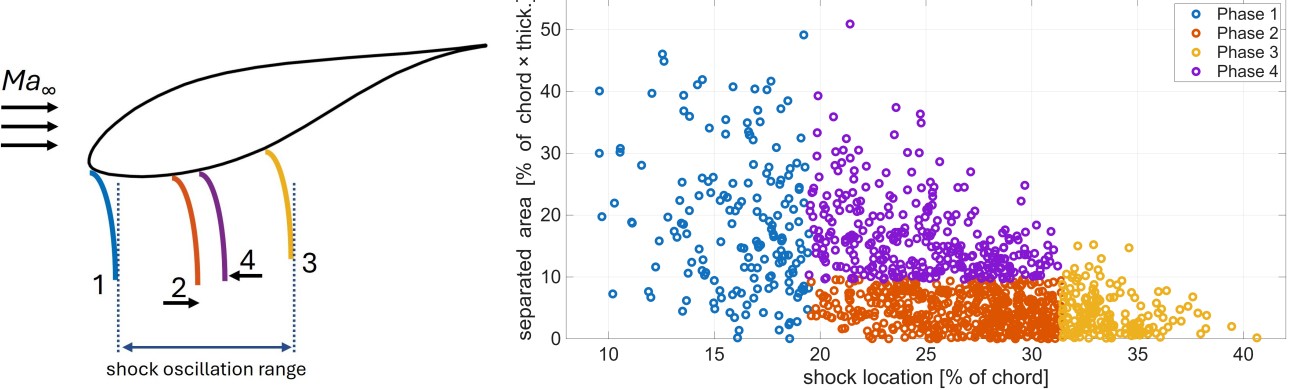

**Figure 17.** Definition of phases (left) and shock locations versus separated areas during various phases for $Ma_\infty = 0.6, AoA = -10°$.

The corresponding phase definitions used in our analysis are illustrated in Fig. 17. Phases 1 and 3 represent the most upstream and most downstream shock positions, respectively, based on a certain threshold. Phases 2 and 4 represent the shock at an intermediate location during its downstream and upstream movement, respectively. A distinction between these two latter phases is made by considering whether the instantaneous separation area is above or below a threshold. The four phases help to represent the unsteadiness in the shock-separation interaction more clearly, which the mean flow-field representation does not capture.

Applying the phase definitions to the PIV dataset yields the phase-averaged flow-fields. The normalized phase-averaged streamwise velocity fields are shown in Fig. 18. The separation area increases when the shock is located most upstream (Phase 1, with the smallest supersonic region) or during upstream motion (Phase 4). When the supersonic region is largest (Phase 3, shock most downstream), the separated flow is more limited. The phase-averaged flow-fields highlight substantial variations in features such as supersonic and separated flow during the transonic buffet cycle. These variations are directly linked to the integral loads experienced by the airfoil, namely lift and drag (D'Aguanno et al., 2025). Although load calculation lies beyond the scope of this study, the extent of supersonic and separated flow across different phases of the buffet cycle can be estimated by evaluating the corresponding flow areas.

In Table 5, the areas of supersonic and separated flow normalized by the square of the chord length for each of the four phases and the overall mean flow-field are presented. In the same table, the relative change in these areas with respect to the mean is presented (in brackets). The table summarizes the visual observations in Fig. 18, and reiterates the impact of shock unsteadiness, which the mean flow-field (Fig. 7e) fails to account for. Another interesting observation is that the separated

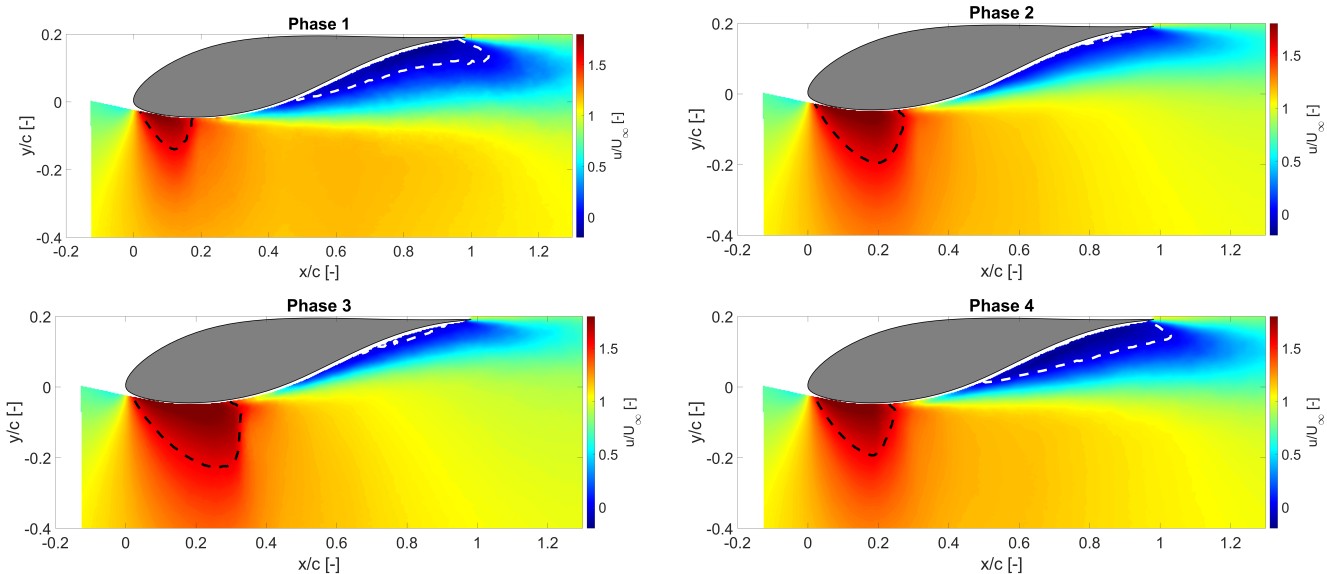

**Figure 18.** Phase-averaged normalized streamwise velocities for $\mathrm{Ma}_\infty = 0.6, \mathrm{AoA} = -10°$. The dashed black lines enclose the local supersonic region, and the white dashed lines enclose the separated flow.

**Table 5.** Supersonic and separation areas during different phases of the transonic buffet cycle.

|  | Supersonic area (change) [% of $c \cdot t_{max}$] | Separation area (change) [% of $c \cdot t_{max}$] | Number of frames (%) |
|---|---|---|---|
| Mean | 13.62 (–) | 9.95 (–) | 1200 (100) |
| Phase 1 | 6.28 (-54%) | 18.76 (+88%) | 171 (14.25) |
| Phase 2 | 14.57 (+7%) | 4.47 (-55%) | 488 (40.67) |
| Phase 3 | 20.86 (+53%) | 4.14 (-58%) | 171 (14.25) |
| Phase 4 | 13.24 (-3%) | 16.38 (+65%) | 310 (25.83) |

flow areas in Phases 1 and 4 (18.76% and 16.38% of $c \cdot t_{max}$) are comparable to the area of separation in the mean flow for $\mathrm{Ma}_\infty = 0.5, \mathrm{AoA} = -10°$ (15.90% of $c \cdot t_{max}$, Fig. 6e). Thus, it is the contribution of Phases 2 and 3 (where flow separation is low) that negates the large separation in Phases 1 and 4 to bring the area of separation in the mean flow down to 9.95% of $c \cdot t_{max}$ for $\mathrm{Ma}_\infty = 0.6, \mathrm{AoA} = -10°$.

In Fig. 19, the areas of supersonic and separated flows in each of the phase-averaged flow-fields are presented, normalized by the corresponding area for the mean flow. Here, the extent of separated flow is seen to range from 0.4 (Phase 3) to 1.9 (Phase 1) times the value in the mean flow. Since extensive flow separation directly contributes to an increase in drag and a decrease in lift over the airfoil, relying solely on the mean flow-field would lead to significantly different load characteristics compared to the instantaneous values. Similarly, owing to the shock-separation interaction, the phase-averaged area of supersonic flow
also ranges from 0.4 (Phase 1) to 1.5 (Phase 3) times its value in the mean flow.

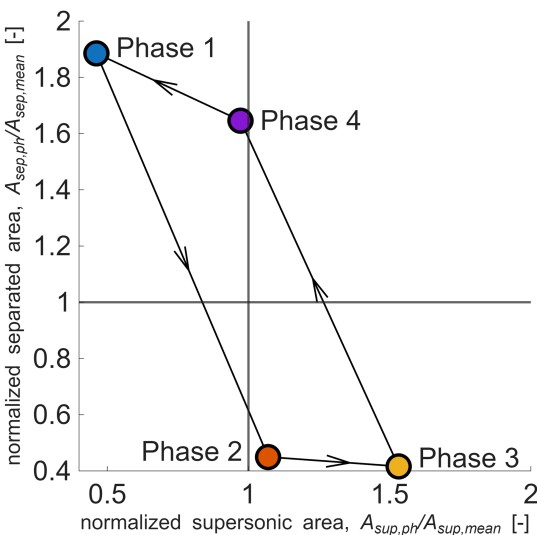

**Figure 19.** Representation of the transonic buffet cycle in terms of phase-averaged supersonic area ($A_{sup,ph}$, along x-axis) and separated area ($A_{sep,ph}$, along y-axis), each normalized with the corresponding area in the mean flow for $Ma_\infty = 0.6$, $AoA = -10°$. The areas occupied in the mean flow by supersonic flow ($A_{sup,mean}$) and separated flow ($A_{sep,mean}$) are denoted by solid black vertical and horizontal lines, respectively.

It is essential to reiterate that the current phase-averaging procedure does not independently reveal the physics of the shock-separation interaction, particularly the variation in separation area during different phases of the shock motion. Instead, the phases already have this asymmetry inherently built into them through the way we choose to define them, as informed by the high-speed schlieren visualization. While the current phase representation provides a convenient approximation to describe the unsteady flow-field, the actual physics is only captured properly by fully resolving the cycle of shock motion through a time-resolved measurement.

## 5   Discussion

As discussed earlier, experimental measurements agree with the predictions of transonic flow occurrence by the envelope in Fig. 3. Experiments also highlight intricacies that are not represented by the transonic envelope. E.g., for $AoA = -10°$, both $Ma_\infty = 0.5 \ \& \ 0.6$ are predicted to lie in the transonic regime. However, the resulting flow-fields are significantly different, with the former experiencing local supersonic flow only intermittently with no shocks, whereas the latter encounters large-amplitude shock-separation interaction. These cases further highlight the need to expand the transonic envelope concept to account for the occurrence of shock waves.

In typical studies of transonic buffet conducted on supercritical airfoils, the shock strength is sufficient to result in separation of the boundary layer downstream. As described by Pearcey et al. (1968), the separation can either be confined to a bubble

close to the shock foot without resulting in trailing-edge separation (Model A), or there can be both a separation bubble and trailing-edge separation present (Model B).

In the present study, at $\text{Ma}_\infty = 0.6$ and AoAs of $-4°$ and $-6°$, oscillating shock waves were captured with both schlieren visualization and PIV. However, there was no significant separation triggered by the shock wave. Moreover, the inclination (AoA) was also not steep enough to result in incipient trailing-edge separation. Thus, no shock-separation interaction, and henceforth, no transonic buffet was encountered in these cases. In other words, the buffet boundary is not crossed in these conditions.

For $\text{Ma}_\infty = 0.6$, $\text{AoA} = -10°$, an oscillating shock wave was present along with significant trailing-edge separation induced by the high incidence, which resulted in transonic buffet. This particular case is categorized as a variant of the Model B interaction defined by Pearcey et al. (1968) and Lee (2001), where trailing-edge separation is already present, and then interacts with the shock. As a result, a typical buffet cycle with large variations in separated flow synchronized with the shock motion is observed in this case.

Some URANS results are presented here to compare with the experimental results. The simulations have been carried out assuming fully turbulent flow, with the $k - \omega$ eddy-viscosity model as turbulence closure. Thus, the boundary layer on the airfoil in the URANS simulations is already turbulent. This is the first aspect of difference with the experiments, which features free boundary layer transition. Secondly, there is a slight mismatch in the Reynolds number. The URANS simulations are conducted at a Reynolds number of $1.8 \times 10^6$, whereas the experiments are carried out at Reynolds numbers of $1.4$ and $1.6 \times 10^6$, corresponding to $\text{Ma}_\infty = 0.5$ and $0.6$, respectively. Additional uncertainties affect $\text{Ma}_\infty$ and AoA in the experiments, in view of blockage and wall interference effects.

Given the differences outlined above, only a qualitative comparison is made between the measured and simulated flow-fields, on the basis of the mean streamwise velocity fields, as shown in Fig. 20, which all apply to AoA$= -6°$. As a first observation, both experiments and URANS predict the emergence of shock waves when increasing $\text{Ma}_\infty$ from $0.5$ to $0.6$. For the experiments, this was already established in previous discussions. For URANS, this is clearly observed when comparing Figures 20b and 20d.

At $\text{Ma}_\infty = 0.5$, the flow-fields share similar features for the experiments and the URANS simulations, as seen in Figs. 20a and 20b. Neither exhibits local supersonic flow. There is some difference in the extent of the accelerated flow regions and the wake, both being more extensive in the URANS simulation.

Upon analyzing the flow fields at $\text{Ma}_\infty = 0.6$ in Figs. 20c and 20d, more pronounced differences between the experiments and URANS emerge. Firstly, the local supersonic pocket (black dotted line) captured in the measurements has a smooth downstream edge, whereas the URANS captures a sharp vertical downstream edge at $x/c \sim 0.35$, indicating a shock wave. In the experiments, an oscillating shock wave was captured, which leads to a smoothing effect in the mean flow representation, as observed here. In this case, the URANS solution only predicts a steady shock. Furthermore, immediately downstream of the local supersonic pocket, the results show two successive regions of flow acceleration near the airfoil surface (around $x/c \sim 0.4$ and $0.5$). These features are not physical but numerical artifacts. Such observations emphasize the limitations of URANS in capturing the correct flow physics.

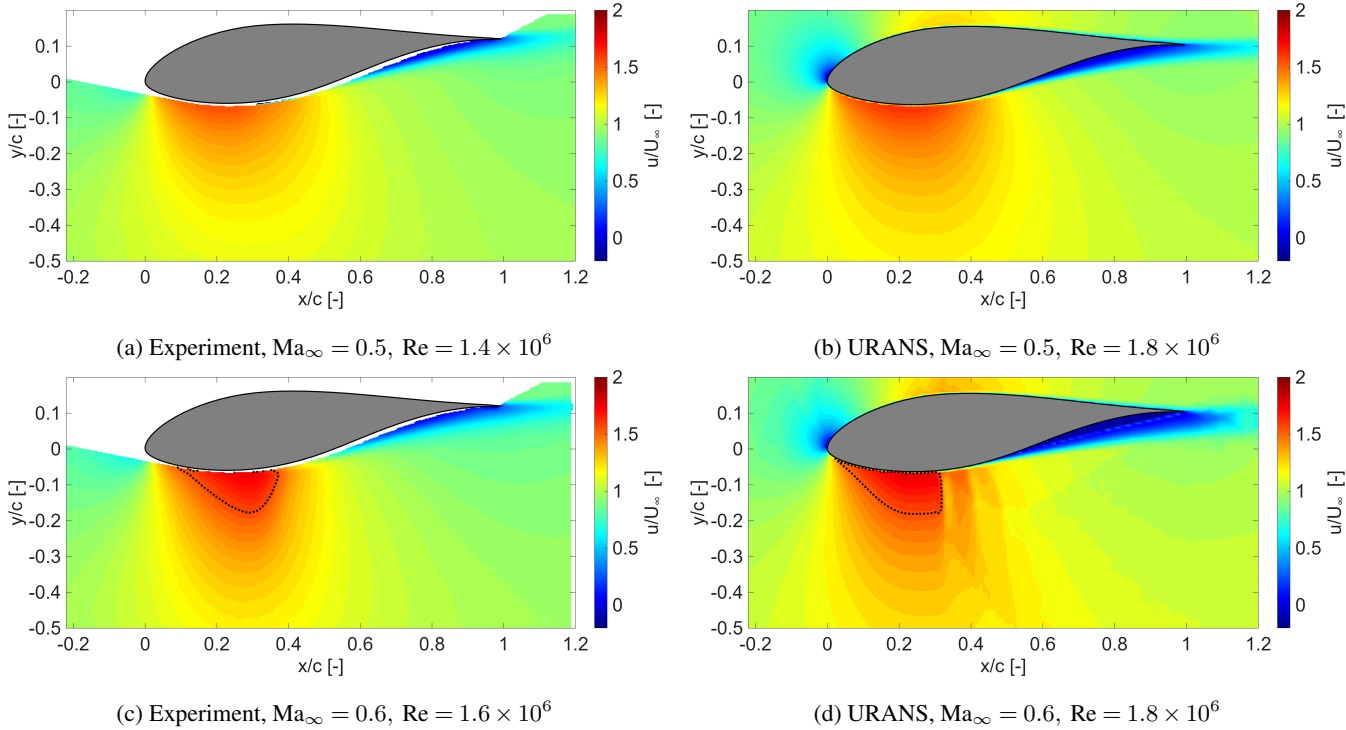

**Figure 20.** Mean normalized streamwise velocity fields from (left column) PIV measurements and (right column) URANS simulations for $\mathrm{Ma}_\infty = 0.5$ (top row) and $\mathrm{Ma}_\infty = 0.6$ (bottom row), at AoA$= -6°$. A local Mach number of 1 is marked with a black dotted line.

The experiments yield instantaneous flow-fields through direct measurements. However, practical constraints do not allow a perfect match in operating conditions (Re, $\mathrm{Ma}_\infty$, and AoA) that are experienced by large wind turbines. In contrast, URANS simulations only capture the mean flow-field and can replicate realistic operating conditions expected at large rotor tips. In the range of conditions where direct comparison is possible, simulations and experiments show agreement in terms of whether or not shock waves occur for the given triplet of $\mathrm{Ma}_\infty$, Re and AoA. However, the URANS approach falls short in modelling the unsteady nature of the shock buffet. Therefore, supporting experiments are deemed necessary to investigate correct unsteady dynamics, e.g., the amplitude and frequency of the shock motion. Consequently, these findings are crucial to inform and validate URANS simulations as well as other numerical techniques that attempt to model the correct physics.

## 6    Conclusions

In this study, experiments were conducted to characterize transonic flow physics over a free-transition (clean) model of the FFA-W3-211 wind turbine airfoil. Wind turbine airfoils feature high thickness and camber, unlike typical supercritical airfoils used for transonic flow studies for aviation applications. Moreover, the conditions at the tips of large wind turbines are also unique, featuring relatively low subsonic inflow Mach number ($\sim 0.3$), high Reynolds number ($\sim 10^7$), and steep, negative

angles of attack. The unique geometry and operating conditions highlight an unexplored problem in the realm of transonic flows.

As shown by Vitulano et al. (2025a), to reproduce the exact transonic flow conditions expected at the blade tips of large wind turbines, similarity in Reynolds number ($\sim 10^7$), inflow Mach number ($\sim 0.3$), and angle of attack is required. Achieving similarity in the first two parameters simultaneously in most, if not all, academically available wind tunnels is not feasible. Since sufficiently high Reynolds numbers to generate shock waves cannot be reached in the current wind tunnel, measurements were instead carried out at higher inflow Mach numbers ($0.5 - 0.6$) to compensate for the lower Reynolds number ($\sim 10^6$). In this way, informed by numerical predictions, the quivalent physics of shock occurrence were investigated.

The current experiments showed good agreement with XFoil and URANS-based calculations for predicting the transition from fully subsonic to transonic flow, either by increasing the inflow Mach number or by steepening the airfoil incidence. The measurements also allowed for characterizing the unsteady nature and dynamics of the shock waves, which had not been captured by other simulation methods previously utilized (XFoil, URANS).

An important finding is that, for the cases studied, transonic buffet was observed only for $\mathrm{Ma}_\infty = 0.6$, $\mathrm{AoA} = -10°$. The interaction of the unsteady shock waves with the separated flow in the wind turbine buffet phenomenon was attributed to the already present trailing-edge separation at the steep inclination. This mechanism is markedly distinct from transonic buffet occurrence studied on supercritical airfoils at low incidences and higher inflow Mach numbers, where the shock is strong enough to trigger trailing-edge separation by itself.

The unsteadiness in the flow-field was characterized by strong periodic variations in the separation region, in sync with the motion of the shock wave. High-speed schlieren imaging revealed that flow separation decreased as the shock wave traversed downstream and increased as the shock wave moved toward the leading edge. Based on this information, PIV data were phase-averaged, and the variations in separated and supersonic flow regions were quantified, demonstrating significant variations compared to the mean. Although unsteady shock waves were observed intermittently at the same $\mathrm{Ma}_\infty$ for less steep inclinations, there was no large-scale separation either triggered or already present in these cases. Furthermore, there was no meaningful correlation between separation extent and shock position. Consequently, no transonic buffet occurred.

The present findings highlight the possibility of inherent unsteadiness associated with the occurrence of shock waves in transonic flow on wind turbine airfoils – even for steady operating conditions. The same unsteady nature will also be reflected in the loads experienced by the airfoil. The characteristic frequencies related to the shock oscillation and subsequent shock-separation interaction need further investigation. If these were to be close to the frequencies of the structural modes, there would be a risk of resonance and increased fatigue loading. In real-world conditions, the inflow and airfoil inclination are always unsteady. Moreover, it is important to bear in mind that this exploratory study was limited to a two-dimensional airfoil section. It is not clear whether (or how) the observed transonic flow effects would materialize on a full-scale blade, where three-dimensional effects, as well as variations of airfoils and flow conditions along the blade and in the inflow, may affect the root causes for transonic flow and for buffeting. More research in this novel field is certainly needed.

*Data availability.* The processed PIV data and post-processing code are openly accessible here: doi.org/10.4121/fbf1c251-cbf9-49d7-9626-a9fe3498aed5.

(NOTE: This link will become active after the paper is accepted for publication.)

## Appendix A: Shock Detection Methodology

While visualizing shock waves qualitatively is relatively straightforward with Schlieren and instantaneous PIV frames, detecting them quantitatively is more demanding. In this section, the methodology developed to detect shock wave locations from instantaneous PIV measurements is discussed. The underlying assumption is that the shock wave is always normal to the freestream direction, which is a reasonable estimation given that only slight deviations are occasionally observed. The sequence of operations used to confirm the presence of a shock and, if present, detect its location in each PIV frame is listed below:

1. Detect the points of maximum gradient in streamwise velocity in a specified region of the PIV frame. All subsequent operations are carried out with reference to these points.

2. From the points detected in the previous operation, eliminate the points that do not have any supersonic flow vectors close upstream or have supersonic flow close downstream.

3. Perform a zeroth-order fit on (i.e., find the mean of) the streamwise locations ($x$ locations, in this case) of the remaining points.

4. Remove the points that are 1.5 standard deviations (chosen based on trial-and-error) away from the mean calculated above, to reinforce the normal shock orientation assumption.

5. If the standard deviation of the remaining points is beyond a specified threshold, or the number of remaining points is beneath a specified limit, then reject the case (i.e., no shock detected).

Shock waves are characterized by strong gradients in the flow-field, but additional constraints need to be applied to ascertain their location. This includes checking the presence of supersonic flow upstream and subsonic flow downstream, to ensure that the detection of points of highest gradients (in velocity, in this case) corresponds to the expected location of the shock front. In cases when the appearance of shocks is intermittent and supersonic flow can sometimes gradually decelerate without resulting in a shock, the additional filtering using the standard deviation in steps 4 and 5 helps in avoiding erroneously observing a shock when there is none.

Some examples of the shock detection methodology in action are presented next. Note that the direction of flow is from left to right. In Fig. A1, the leftmost figure shows the points (as magenta squares) that have the maximum gradient in streamwise velocity at respective transverse locations. A few points towards the bottom lie inside the supersonic flow pocket (marked with a solid black line) rather than on its downstream edge, as expected for a shock front. As seen in the middle frame, these points are eliminated after checking for supersonic flow upstream (which holds true) and subsonic flow downstream (which is violated).

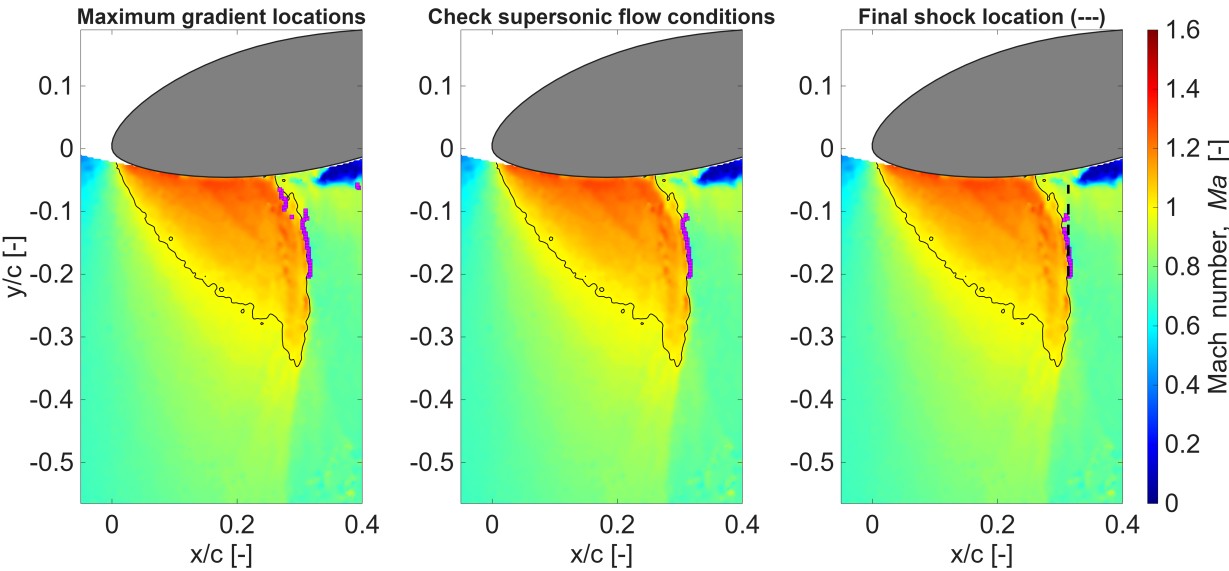

**Figure A1.** Instantaneous local Mach number field for $\mathrm{Ma}_\infty = 0.6$, AoA$= -10°$ with local supersonic flow pocket (marked with solid black line), detected shock locations (magenta markers) and the final shock front location (dashed black line) on the right.

Finally, as seen in the right frame, the final detected shock location is marked with a dashed black line after performing the checks regarding the standard deviation on the remaining points.

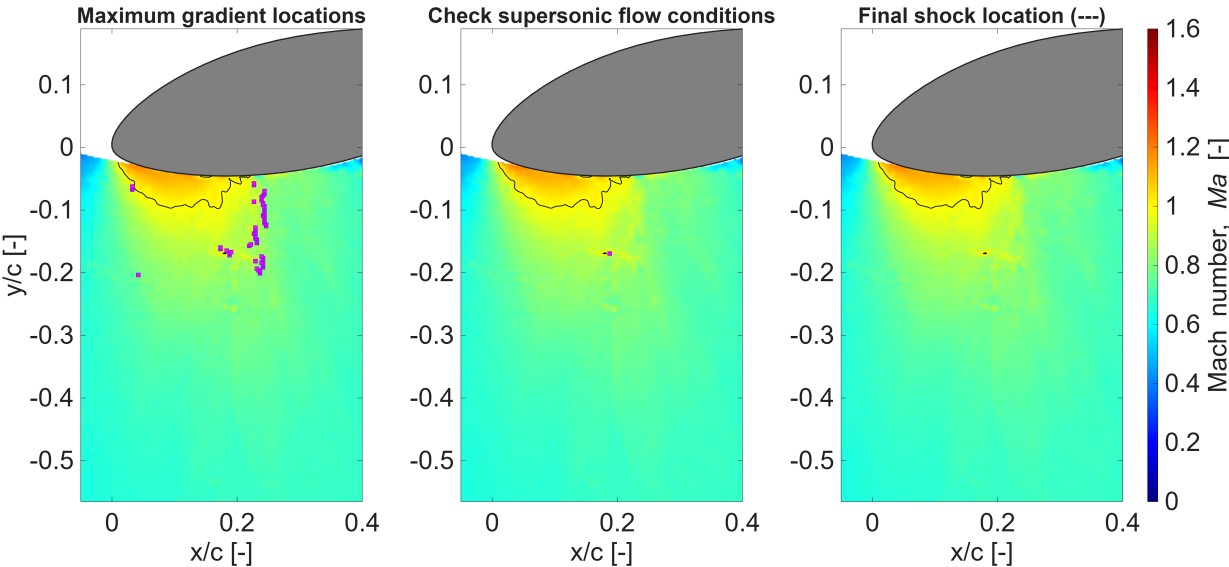

**Figure A2.** Instantaneous local Mach number field for $\mathrm{Ma}_\infty = 0.6$, AoA$= -10°$ with local supersonic flow pocket (marked with solid black line) and detected shock locations (magenta markers). No shock front is detected after the filtering process, as seen on the right.

Another example is presented in Fig. A2, where it is already expected that no shock wave is present. The supersonic up-
stream/subsonic downstream check (shown in the middle frame) takes care of almost all the potential shock front points
detected by the maximum streamwise velocity gradient (left), except for a few next to a tiny supersonic pocket at $x/c \approx 0.2$.
The next filtering step, based on the number of remaining points and the standard deviation, eliminates the last few detected
points as well, rightly resulting in no shock front detected for the case.

## Appendix B: Additional Schlieren Results

A full set of schlieren images is shown here for all combinations of $Ma_\infty = 0.5, 0.55, 0.6$ and AoAs$= -4°, -6°, -10°$. These
are instantaneous snapshots of the flow-field. Shock waves appear in select configurations: Figs. B1 (c),(e),(f),(h),(i). denoted
by thick, dark gray lines extending vertically down from the bottom surface of the airfoil.

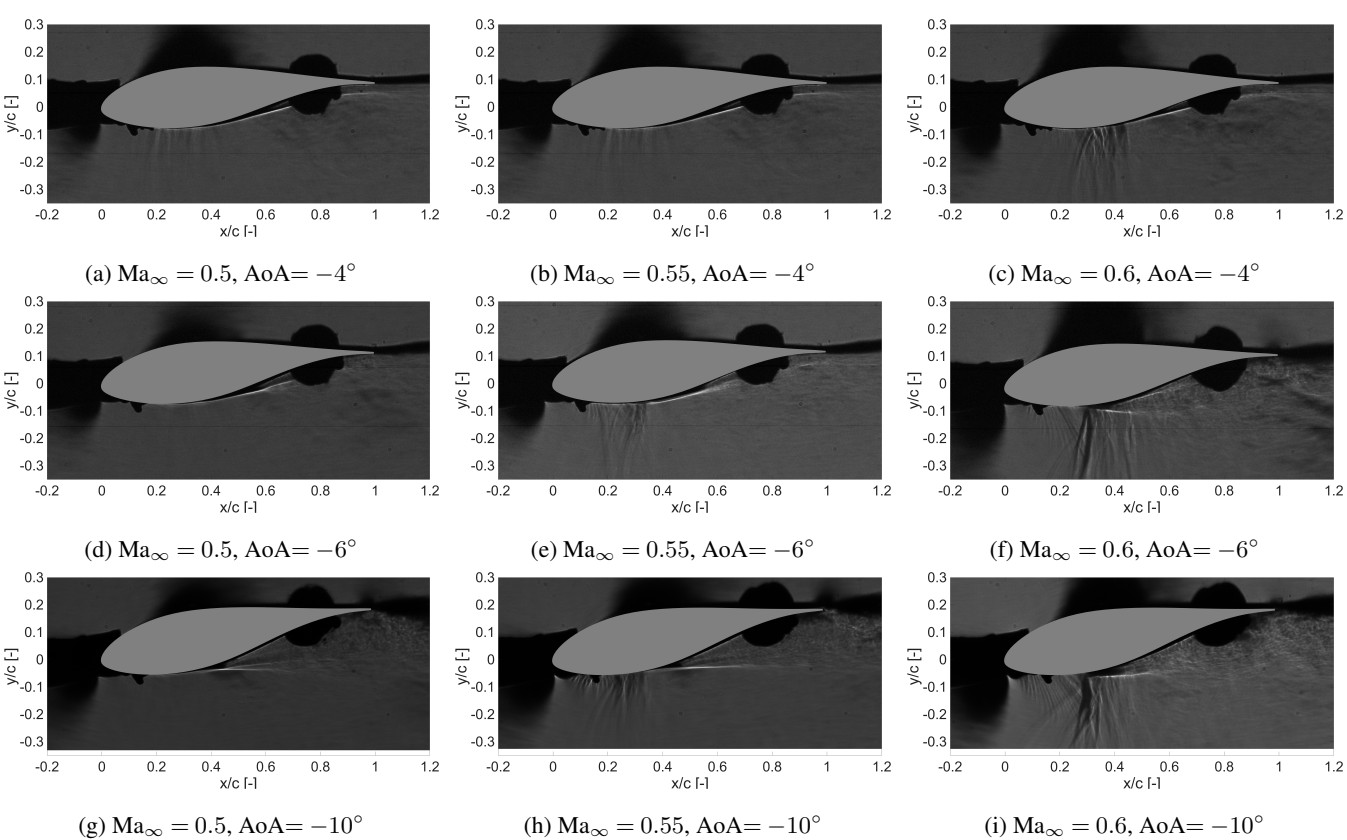

**Figure B1.** Instantaneous Schlieren images showing the appearance of shock waves in (c), (e), (f), (h), and (i).

## Appendix C: Additional shock location versus separation area plots

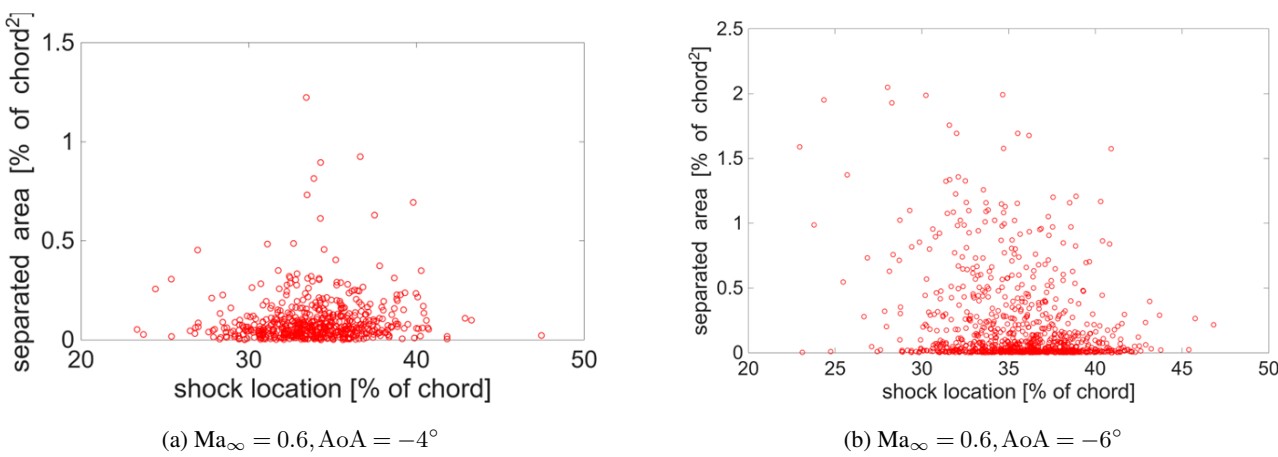

(a) $\mathrm{Ma}_\infty = 0.6, \mathrm{AoA} = -4°$           (b) $\mathrm{Ma}_\infty = 0.6, \mathrm{AoA} = -6°$

**Figure C1.** Shock location versus separation area, demonstrating that neither case has a clear correlation between the two.

*Author contributions.* AA was responsible for the overall research, in collaboration with MCV and under the supervision of DDT, FS, BvO, and DvT. AA carried out the experimental measurements as well as the post-processing of the measurements with the support of FS and BvO. The results were visualized by AA and analyzed by AA, DDT, FS, BvO, and DvT. MCV conducted simulations to generate the underlying data of Fig. 3 and the URANS data in Fig. 20. The first draft was prepared by AA, and subsequent corrections were made according to the reviews by MCV, DDT, FS, BvO, and DvT.

*Competing interests.* The authors declare that they have no conflict of interest.

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
