# Peer review of "Experimental study of transonic flow over a wind turbine airfoil"

_Wind Energy Science, 2025_

## Author Comment (AC1)

**Response to RC1**

July 11, 2025

We understand the challenge of the handling editor in finding reviewers for our manuscript on transonic flows in the wind energy community and, therefore, truly appreciate the efforts of the reviewers. Reviewer 1 had strong reservations to publish this manuscript based on, in particular, claims of a lack of (1) knowledge of the authors, (2) novelty of the work, and (3) relevance to the wind energy science community. In addition, some further comments were made and questions were raised. We strongly object to the main claims above, as explained in the following. Nevertheless, considering the feedback, we will make changes accordingly to improve our manuscript.

- **Knowledge of the authors**

  All authors were vested in the work carried out together and, among them, have published more than 25 scientific contributions on the topics of transonic buffet and supersonic flows including the research highlight in a leading journal in the field. This reflects decades of experience in the field, which a simple Google Scholar (or Scopus) search would substantiate. The claim of the reviewer of "a lack of knowledge on part of the authors in both compressible flow and scientific literature on transonic flows with respect to shock boundary-layer interactions (SBLI) and the physics of buffeting" is therefore surprising to us.

- **Novelty of the work**

  Transonic buffet is observed in a variety of applications and has been widely studied in aviation, where it is recognized as being an important phenomenon relevant to the operational performance. Unlike aviation airfoils, wind turbine tip airfoils are characterised by a much larger thickness-to-chord ratio and also high camber. Moreover, the buffet phenomenon on wind turbine airfoils is expected to occur at high Reynolds number $Re$ (of the order of $10^7$, similar to aviation), but relatively low subsonic free-stream Mach number $Ma$ in combination with steeper inclinations in the opposite direction (i.e., large negative angle of attack) compared to aviation applications (De Tavernier & von Terzi, 2022). Hence, these distinct conditions under which buffeting may develop (aviation vs wind turbines) likely imply that the knowledge of one area may not be directly transferable to the other. This makes it very relevant to dedicate studies to the typical characteristics of transonic buffet produced on typical wind turbine airfoils, given the uniqueness of the geometry and operational conditions.

  An example highlighting the difference in the transonic buffet cycle produced on a supercritical airfoil for aviation applications and the wind turbine airfoil investigated in the present manuscript is shown in Fig. 1. The probability density function (pdf) of the shock location as percentage of chord for the supercritical airfoil is flatter, with a higher density towards the ends of the range of motion whereas, on the wind turbine airfoil, the density is higher towards the center of the range of motion.

  To the knowledge of the authors, this is the first time that transonic flow on any wind turbine airfoil was studied experimentally. Preceding studies had been either theoretical or numerical (URANS). The simulation tools applied are inherently limited in modeling and predicting the associated dynamics of shock occurrence, such as the amplitude and frequency of the buffet. Thus, experiments are crucial not only to investigate the physics but also to correctly model the dynamics using lower-order tools and to provide validation data.

[Figure]

[Figure]

Figure 1: Probability density function of the shock location as a percentage of chord for transonic buffet. Left: OAT-15A supercritical airfoil, from D'Aguanno et al. (2021); right: FFA-W3-211 wind turbine airfoil (current study).

Contrary to the claim of the reviewer, we consider our current contribution as novel to both wind energy science and the literature on transonic flows. To support this, we will extend the introduction of the revised manuscript to provide more context on the differences in transonic buffet between aviation and wind energy applications so that the novelty can be appreciated more easily. We will include a reference to the review paper of Giannelis et al. (2017) for more literature on the aviation application.

In addition, we will expand the results section of the revised manuscript to describe the flow dynamics uncovered through our experiments in more detail. To this end, we are using a phase-averaged analysis of the flowfield based upon the shock location that builds towards the graph shown in Fig. 1. This will help address the concerns regarding both novelty in general and relevance for wind energy applications. A preview of these results is shown in Fig. 2, where the phase-averaged representation of the buffet cycle reveals the marked changes in the flowfield resulting from the motion of the shock.

[Figure]

Figure 2: Phase-averaged (based on shock location) streamwise velocity fields for $Ma_\infty = 0.6$, AoA$=-10°$ of the FFA-W3-211 wind turbine airfoil (current study); black and white dashed lines identify supersonic and separated flow regions, respectively.

- **Relevance to the wind energy community**

The investigated airfoil is used at the tip of prominent reference wind turbines, namely the IEA 15MW (see Gaertner et al., 2020) and IEA 22MW (see Zahle et al., 2024). These are widely used for scientific research in the wind energy community. However, there are very few limited datasets on polars for this airfoil, and none with compressibility effects. As the turbine was designed with polars from simulation tools, it is critical to provide insights into physical mechanisms that these tools are unable to capture.

Regarding the choice of $Ma$ and $Re$ values for the experiments. As pointed out in the manuscript, these differ from the above mentioned reference turbines, in order to qualitatively capture the relevant physics expected at full scale. However, perhaps counter to expectation, $Ma$ is not the single parameter relevant for the occurrence of compressibility effects. Vitulano et al. (2025) showed that an increase in $Re$ plays a crucial role in accelerating the appearance of shock waves at relatively lower $Ma$. This is shown in the figure below. For an $Re$ of $1.8 \times 10^6$, which is close to the $Re$ achieved in the current experiments, shock waves occur only at a $Ma$ of 0.6. However, at an increased $Re$ of $9 \times 10^6$, shock waves start to occur already at a $Ma$ of 0.45, close to an angle of attack of -9 deg. Moreover, hysteresis effects on a pitching airfoil result in shock waves being observable at an even lower $Ma$ of 0.35, i.e. close to Mach numbers that can be observed on large turbines like the IEA 22MW reference wind turbine. However, this combination of a very high $Re$ and a relatively high $Ma$ is not possible to reproduce in most, if not all, experimental facilities available for wind energy research. Thus, to produce shock waves on such an airfoil experimentally, the higher $Ma$ in our study is justified due to the limitations of achieving a $Re$ of the order of $10^7$ in the same facility, such that the physics of shock occurrence being investigated in the experiments is the same as expected for a full-scale turbine.

[Figure]

Figure 3: Subsonic-supersonic boundary for the FFA-W3-211 wind turbine tip airfoil, with symbols indicating URANS simulations showing no supersonic flow (grey crosses), supersonic regime established (red circles), and configurations in which shock waves appear (green squares); from Vitulano et al. (2025).

In the original manuscript, we briefly alluded to the argument made above when we mention the study by Vitulano et al. (2025) on line 168. In the revised manuscript, we intend to more clearly and explicitly justify the choice of $Ma$ and $Re$. It is critical to understand that in order to reproduce the same physics of shock occurrence as in full-scale, the required change in $Re$, for any wind tunnel, necessitates to choose the corresponding $Ma$ and pitch angle (see Fig. 3).

**Response to additional comments and questions of the reviewer:**

- The reviewer is wondering why the numerical simulations of Vitulano et al. (2025) are not at the same conditions as in the experiment. In the numerical study, a range of $0.2 \leq Ma \leq 0.7$ and $1.8 \times 10^6 \leq Re \leq 9 \times 10^6$ (see also Fig. 3) were investigated to determine the threshold for transonic flow and shock occurrence. This includes all relevant $Ma$ and bridges from the $Re$ in an incompressible experiment used for validation to the value in full scale. In the article, more details were shown for the full-scale operation conditions due to its practical relevance. For the present experiments, a slightly lower $Re$ of $1.5 \times 10^6$ than for the incompressible experiment was achieved and the $Ma$ and angle of attack (AoA) were chosen to obtain similar physics as expected for full-scale considering the difference in $Re$ as explained above.

- The reviewer suggests that some rather simple design or operational changes would "solve" transonic flow issues. Unfortunately, avoiding transonic flow, if encountered on wind turbines, may not be as straightforward as the reviewer seems to believe. The patent of von Terzi et al. (2023) (and a corresponding research article in review) delves into more detail on this. A key consideration is that the transonic operational boundaries are wind-speed dependent and, with the inherent variability of the wind resource, it is possible to end up in a challenging situation where escaping transonic flow may not be possible or, at least, the right combination of tip speed and pitch needs to be known to exit safely. Other suggestions will lead to a lower energy yield and increase the cost of energy. With a good physical understanding, it is, however, possible to navigate the "mine field" with minimal performance losses or by simply designing a better turbine pushing the boundary of transonic flow out of the operational range. Thus, it remains crucial to acquire reliable experimental data on the behaviour of wind turbine airfoils under transonic conditions.

- Regarding the use of RFOIL, we appreciate the trust of the reviewer in tools (co-)developed in our institute. Here, however, a choice was made to provide a freely accessible tool (XFOIL) for better reproducibility. While we also believe in some benefits of RFOIL, there are many versions available and no access to source code is given. Moreover, the fundamental limitations of the approach for investigating transonic flow, whether XFOIL or RFOIL, remain.

- Regarding the comments on a missed opportunity for verification and validation of the current experiments with numerical simulations from Vitulano et al. (2025), a separate publication is in preparation that stands apart from the current study (with specific experimental focus) and, in our opinion, should not be crammed together. We plan to openly publish the experimental dataset and invite other groups to carry out V&V of their numerical tools and models with our data. Also, as assumed by the reviewer and suggested in Vitulano et al. (2025), there are plans for the use of Hybrid RANS/LES methods to be applied. These efforts are beyond the scope of the present study and communication.

- Regarding the desire for lift and drag coefficients, this is understandable if the data were to be used directly in other simulation/design tools. Similarly, frequencies induced by the buffeting would be important to know for structural design. However, the present manuscript aimed in establishing qualitatively, and for the first time, the transonic flow behaviour of the wind turbine airfoil to reveal if and when shocks will occur and where they will manifest themselves. The manuscript also provides flow details to serve as a validation for future high-fidelity simulations and reduced order models. Again, the chosen operating parameters are not the same as full-scale, but chosen to match the expected physical behavior. Hence, quantitative analysis on a performance loss are less meaningful, but the qualitative behavior of the flow must be understood or attempts on avoiding buffeting risks will be fortuitous at best.

**Summary**

All previous literature on transonic buffet studies has focused on airfoils with low thickness-to-chord ratio (of the order of 10%), moderate camber, and a vastly different angle-of-attack regime. For the first time, a wind turbine airfoil characterized by a higher thickness-to-chord ratio, high camber and negative inclination as relevant for above-rated wind speeds, has been studied in transonic flow conditions, and the resulting buffet phenomenon was observed to be distinct compared to supercritical (aviation) airfoils. This highlights the novelty of the work.

On the full-scale IEA 15MW and 22MW reference wind turbines and similar-size turbines currently designed in industry, shocks might occur at blade tips with $Re$ of the order of $10^7$ and $Ma \sim 0.3$. However, in the transonic wind tunnel facility, it is only possible to reach $Re$ of the order of $10^6$, which would not be sufficient to produce shocks at $Ma \sim 0.3$. Recall, the appearance of shocks was shown to depend on both $Re$ and $Ma$ by Vitulano et al. (2025). Thus, the experiments were conducted at $Ma$ of 0.5 and above, to study the physics of transonic flow with ($Ma = 0.6$) and without ($Ma = 0.5$) the occurrence of shocks on the wind turbine airfoil.

We thank the reviewer for valuable comments and questions that initiated amendments to the manuscript to address the aforementioned points more clearly. In addition, we will add further results to discuss in more detail the dynamics of the buffet cycle (as shown above). Overall, we want to highlight the distinct features and relevance of the possibility of transonic buffet on wind turbine airfoils. We believe that these modifications will better substantiate the novelty and relevance of our work, such that the (revised) manuscript can be considered as suitable for publication in Wind Energy Science.

**References**

De Tavernier, D., & von Terzi, D. (2022). The emergence of supersonic flow on wind turbines. In *Journal of Physics: Conference Series* (pp. 042068).

D'Aguanno, A., Schrijer, F.F.J., & van Oudheusden, B.W. (2021). Experimental investigation of the transonic buffet cycle on a supercritical airfoil. *Experiments in Fluids*, 62, 1-23.

Giannelis, N.F., Vio, G.A., Levinski O. (2017). A review of recent developments in the understanding of transonic shock buffet. *Prog Aerosp Sci* 92:39–84.

Gaertner, E., Rinker, J., Sethuraman, L., Zahle, F., Anderson, B., Barter, G., Abbas, N., Meng, F., Bortolotti, P., Skrzypinski, W., Scott, G., Feil, R., Bredmose, H., Dykes, K., Shields, M., Allen, C., and Viselli, A. (2020). Definition of the IEA Wind 15-Megawatt Offshore Reference Wind Turbine. *Tech. Rep., National Renewable Energy Laboratory (NREL), Golden, CO*.

Zahle, F., Barlas, T., Lonbaek, K., Bortolotti, P., Zalkind, D., Wang, L., Labuschagne, C., Sethuraman, L., and Barter, G. (2024). Definition of the IEA Wind 22-Megawatt Offshore Reference Wind Turbine. *Tech. rep., National Renewable Energy Laboratory (NREL), Golden, CO (United States)*.

Vitulano, M., De Tavernier, D., De Stefano, G., & von Terzi, D. (2025). Numerical analysis of transonic flow over the FFA-W3-211 wind turbine tip airfoil. *Wind Energy Science*, 10(1), 103–116.

von Terzi, D., De Tavernier, D.& Zaayer, M. (2023). Method of operating a wind turbine. *Patent WO2025127925A1*, international publication date 19.6.2025.

---

## Author Response (AR2)

**Response to reviews: WES-2025-65**
**Experimental study of transonic flow over a wind turbine airfoil**

**November 2025**

The authors would like to thank the referees for their effort and comments that have served to improve this manuscript. The *referees' comments* are sequentially presented. Each *comment* is followed by the authors' response marked in blue, and the subsequent change/addition made to the manuscript marked in red.

**Referee #1**

*1. The experiments appear to be original, and the uncertainty quantification of both the wind tunnel conditions and PIV measurements appear adequate. Overall though, there is no clear scientific contribution that is relevant to the wind energy science community. The experimental conditions chosen (and feasible) in the TST-27 wind tunnel are paradoxically disconnected from previous studies of the same research group on the general scaling of tentative transonic conditions in Region III [de Tavernier v. Terzi, 2022] and recent numerical investigations of transonic flow around the same airfoil [Vitulano et al., WES 2025]. Similarly, the discussion of observations with respect to shock formation reveals a lack of knowledge on part of the authors in both compressible flow and scientific literature on transonic flows with respect to shock boundary-layer interactions (SBLI) and the physics of buffeting. Consequently, the paper is neither a contribution to wind energy science nor to the literature on transonic flows.*

Reviewer 1 had strong reservations about publishing this manuscript based on, in particular, claims of a lack of (1) knowledge of the authors, (2) novelty of the work, and (3) relevance to the wind energy science community. These points are addressed below.

- **Knowledge of the authors**

  All authors were vested in the work carried out together and, among them, have published more than 25 scientific contributions on the topics of transonic buffet and supersonic flows, including the research highlight in a leading journal in the field. This reflects decades of experience in the field, which a simple Google Scholar (or Scopus) search would substantiate. The claim of the reviewer of "a lack of knowledge on part of the authors in both compressible flow and scientific literature on transonic flows with respect to shock boundary-layer interactions (SBLI) and the physics of buffeting" is therefore surprising to us.

- **Novelty of the work**

  Transonic buffet is observed in a variety of applications and has been widely studied in aviation, where it is recognized as being an important phenomenon relevant to the operational performance. Unlike aviation airfoils, wind turbine tip airfoils are characterised by a much larger thickness-to-chord ratio and also high camber. Moreover, the buffet phenomenon on wind turbine airfoils is expected to occur at high Reynolds number $Re$ (of the order of $10^7$, similar to aviation), but relatively low subsonic free-stream Mach number $Ma$ in combination with steeper inclinations in the opposite direction (i.e., large negative angle of attack) compared to aviation applications (De Tavernier & von Terzi, 2022). Hence, these distinct conditions under which buffeting may develop (aviation vs wind turbines) likely imply that the knowledge of one area may not be directly transferable to the other. This makes it very relevant to dedicate studies to the typical characteristics of transonic buffet produced on typical wind turbine airfoils, given the uniqueness of the geometry and operational conditions.

[Figure]

[Figure]

Figure 1: Probability density function of the shock location as a percentage of chord for transonic buffet. Left: OAT-15A supercritical airfoil, from D'Aguanno et al. (2021); right: FFA-W3-211 wind turbine airfoil (current study).

An example highlighting the difference in the transonic buffet cycle produced on a supercritical airfoil for aviation applications and the wind turbine airfoil investigated in the present manuscript is shown in Fig. 5. The probability density function (pdf) of the shock location as percentage of chord for the supercritical airfoil is flatter, with a higher density towards the ends of the range of motion whereas, on the wind turbine airfoil, the density is higher towards the center of the range of motion.

To the knowledge of the authors, this is the first time that transonic flow on any wind turbine airfoil was studied experimentally. Preceding studies had been either theoretical or numerical (URANS). The simulation tools applied are inherently limited in modeling and predicting the associated dynamics of shock occurrence, such as the amplitude and frequency of the buffet. Thus, experiments are crucial not only to investigate the physics but also to correctly model the dynamics using lower-order tools and to provide validation data.

Contrary to the claim of the reviewer, we consider our current contribution as novel to both wind energy science and the literature on transonic flows. To support this, we will extend the introduction of the revised manuscript to provide more context on the differences in transonic buffet between aviation and wind energy applications so that the novelty can be appreciated more easily. We will include a reference to the review paper of Giannelis et al. (2017) for more literature on the aviation application.

In addition, we will expand the results section of the revised manuscript to describe the flow dynamics uncovered through our experiments in more detail. To this end, we are using a phase-averaged analysis of the flowfield based upon the shock location that builds towards the graph shown in Fig. 5. This will help address the concerns regarding both novelty in general and relevance for wind energy applications. A preview of these results is shown in Fig. 2, where the phase-averaged representation of the buffet cycle reveals the marked changes in the flowfield resulting from the motion of the shock.

The revised manuscript has an extended introduction on the transonic buffet studied in aviation (l. 49-58), the major differences between wind turbine airfoils and supercritical airfoils (l. 59-63). Additionally, the Results include a new section focused on the shock-separation interaction observed in the current study (Section 4.4, l. 393). Also, the Discussion (section 5) has been extended, featuring more about the transonic buffet observed in our study.

- **Relevance to the wind energy community**

The investigated airfoil is used at the tip of prominent reference wind turbines, namely the IEA 15MW (see Gaertner et al., 2020) and IEA 22MW (see Zahle et al., 2024). These are widely used for scientific research in the wind energy community. However, there are very few limited datasets on polars for this airfoil, and none with compressibility effects. As the turbine was designed with polars from simulation tools, it is critical to provide insights into physical mechanisms that these tools are unable to capture.

Regarding the choice of $Ma$ and $Re$ values for the experiments. As pointed out in the manuscript, these differ from the above mentioned reference turbines, in order to qualitatively capture the relevant physics expected at full scale. However, perhaps counter to expectation, $Ma$ is not the single parameter relevant for the occurrence of compressibility effects. Vitulano et al. (2025a) showed that an increase in $Re$ plays a crucial role in accelerating the appearance of shock waves at relatively lower $Ma$. This is shown in

[Figure]

Figure 2: Phase-averaged (based on shock location) streamwise velocity fields for $Ma_\infty = 0.6$, AoA$=-10°$ of the FFA-W3-211 wind turbine airfoil (current study); black and white dashed lines identify supersonic and separated flow regions, respectively. Also included in the revised manuscript (Fig. 18).

the figure below. For an $Re$ of $1.8 \times 10^6$, which is close to the $Re$ achieved in the current experiments, shock waves occur only at a $Ma$ of 0.6. However, at an increased $Re$ of $9 \times 10^6$, shock waves start to occur already at a $Ma$ of 0.45, close to an angle of attack of -9 deg. Moreover, hysteresis effects on a pitching airfoil result in shock waves being observable at an even lower $Ma$ of 0.35, i.e. close to Mach numbers that can be observed on large turbines like the IEA 22MW reference wind turbine. However, this combination of a very high $Re$ and a relatively high $Ma$ is not possible to reproduce in most, if not all, experimental facilities available for wind energy research. Thus, to produce shock waves on such an airfoil experimentally, the higher $Ma$ in our study is justified due to the limitations of achieving a $Re$ of the order of $10^7$ in the same facility, such that the physics of shock occurrence being investigated in the experiments is the same as expected for a full-scale turbine.

In the original manuscript, we briefly alluded to the argument made above when we mention the study by Vitulano et al. (2025a) on line 168. In the revised manuscript, we intend to more clearly and explicitly justify the choice of $Ma$ and $Re$. It is critical to understand that in order to reproduce the same physics of shock occurrence as in full-scale, the required change in $Re$, for any wind tunnel, necessitates to choose the corresponding $Ma$ and pitch angle (see Fig. 3).

The revised manuscript has an extended discussion on the choice of experimental conditions in Experimental Design (section 2). It includes the $Re$ dependency discussed above, providing more clarity and justification of the choices made in the current study.

*2. It is unclear why recent numerical investigations on the same airfoil by the same research group [Vitulano et al., WES 2025] have not been applied to the experimental conditions investigated in this paper. The reviewer thinks that this is a missed opportunity for an easy-to-do validation study.*

The reviewer is wondering why the numerical simulations of Vitulano et al. (2025a) are not at the same conditions as in the experiment. In the numerical study, a range of $0.2 \leq Ma \leq 0.7$ and $1.8 \times 10^6 \leq Re \leq 9 \times 10^6$ (see also Fig. 4) were investigated to determine the threshold for transonic flow and shock occurrence. This includes all relevant $Ma$ and bridges from the $Re$ in an incompressible experiment used for validation to the value in full scale. In the article, more details were shown for the full-scale operation conditions due to its practical relevance. For the present experiments, a slightly lower $Re$ of $1.5 \times 10^6$ than for the incompressible experiment was achieved and the $Ma$ and angle of attack (AoA) were chosen to obtain similar physics as

[Figure]

**(a) Re** $= 1.8 \times 10^6$        **(c) Re** $= 9 \times 10^6$

Figure 3: Subsonic-supersonic boundary for the FFA-W3-211 wind turbine tip airfoil, with symbols indicating URANS simulations showing no supersonic flow (grey crosses), supersonic regime established (red circles), and configurations in which shock waves appear (green squares); from Vitulano et al. (2025a). Also included in revised manuscript (Fig. 2).

expected for full-scale, considering the difference in $Re$ as explained above.

The revised manuscript includes further justification on the experimental design of the current study in section 2, and also includes a qualitative comparison between experimental measurements and URANS simulations (Fig. 20) in the Discussion (section 5).

*3. The reviewer very much appreciates the TORQUE 2022 paper by the same research group [de Tavernier & v. Terzi, 2022] that brought attention to the possibility of transonic flow in high-speed Region III where blade pitch control results in order -10deg angle of attack near the blade tip. In this regard, Fig. 3d of [de Tavernier & v. Terzi, 2022] has both all relevant information and actually also the solution to the problem with respect to blade design (e.g. reduce tip chord to increase angle of attack, or lower both tip speed and blade pitch to achieve the same rated power at higher angle of attack). This conclusion is not changing for the IEA-22MW turbine as the associated increase in tip speed compared to the IEA-15MW turbine is not going to ever result in Mach numbers higher than 0.5. Therefore, the TST-27 is simply not the best wind tunnel to conduct relevant experiments. In other words, the experimental conditions chosen (see Fig. 4 of present paper) are simply not relevant.*

The reviewer suggests that some rather simple design or operational changes would "solve" transonic flow issues. Unfortunately, avoiding transonic flow, if encountered on wind turbines, may not be as straightforward as the reviewer seems to believe. The patent of von Terzi et al. (2023) (and a corresponding research article in review) delves into more detail on this. A key consideration is that the transonic operational boundaries are wind-speed dependent, and, with the inherent variability of the wind resource, it is possible to end up in a challenging situation where escaping transonic flow may not be possible or, at least, the right combination of tip speed and pitch needs to be known to exit safely. Other suggestions will lead to a lower energy yield and increase the cost of energy. With a good physical understanding, it is, however, possible to navigate the "mine field" with minimal performance losses or by simply designing a better turbine pushing the boundary of transonic flow out of the operational range. Thus, it remains crucial to acquire reliable experimental data on the behaviour of wind turbine airfoils under transonic conditions.

The relevance of the chosen inflow Mach numbers in the current study is elaborated upon in the revised Experimental Design (section 2), explaining how the results of Vitulano et al. (2025a) highlight the Reynolds number dependency of the occurrence of shock waves, making it impossible to reproduce them at low inflow Mach numbers in the current wind tunnel setup (TST-27).

*4. Also, Xfoil is not suitable for thick airfoils. This is known and was one of the objectives for TU Delft to develop Rfoil. Why are TU Delft authors not using Rfoil ?*

Regarding the use of RFOIL, we appreciate the trust of the reviewer in tools (co-)developed in our institute. Here, however, a choice was made to provide a freely accessible tool (XFOIL) for better reproducibility. While we also believe in some benefits of RFOIL, there are many versions available and no access to source code is given. Moreover, the fundamental limitations of the approach for investigating transonic flow, whether XFOIL or RFOIL, remain.

*5. Missed V&V opportunities. As mentioned above, it is curious why the same research group would not conduct numerical simulations (RANS, URANS, DDES, steady and unsteady) of the experimental conditions considered in this paper and benchmarked against research in the compressible flow community.*

Regarding the comments on a missed opportunity for verification and validation of the current experiments with numerical simulations from Vitulano et al. (2025a), a separate publication is in preparation that stands apart from the current study (with specific experimental focus) and, in our opinion, should not be crammed together. We plan to openly publish the experimental dataset and invite other groups to carry out V&V of their numerical tools and models with our data. Also, as assumed by the reviewer and suggested in Vitulano et al. (2025a), there are plans for the use of Hybrid RANS/LES methods to be applied. These efforts are beyond the scope of the present study and communication.
The revised manuscript briefly discusses URANS simulations compared to experimental results in Discussion (section 5, Fig. 20) to highlight qualitative agreements and differences between the two.

*6. The lack of lift and drag data also makes the present work less relevant as the drag rise and associated loss in airfoil cl/cd would have a quantified effect on tentative performance loss. Once more, Ma > 0.5 are not relevant to utility-scale turbines, and transonic flow conditions can be avoided both by rotor design and operation in high-speed Region III.*

Regarding the desire for lift and drag coefficients, this is understandable if the data were to be used directly in other simulation/design tools. Similarly, frequencies induced by the buffeting would be important to know for structural design. However, the present manuscript aimed in establishing qualitatively, and for the first time, the transonic flow behaviour of the wind turbine airfoil to reveal if and when shocks will occur and where they will manifest themselves. The manuscript also provides flow details to serve as a validation for future high-fidelity simulations and reduced order models. Again, the chosen operating parameters are not the same as full-scale, but chosen to match the expected physical behavior. Hence, quantitative analysis on a performance loss are less meaningful, but the qualitative behavior of the flow must be understood or attempts on avoiding buffeting risks will be fortuitous at best.

*7. There is a wealth of literature available on shock boundary-layer interaction and shock buffeting. The reviewer only sees one reference to work in this area. This is unfortunate as it leaves the present work without contribution in that area.*

The revised manuscript will include more background on transonic buffet in the introduction and results.
Extensive background and discussion on transonic buffet in the revised Introduction (section 1), Results (section 4), Discussion (section 5). Consequently, ten additional references from transonic flows have been cited in the revised manuscript.

We thank the reviewer for valuable comments and questions that initiated amendments to the manuscript to address the aforementioned points more clearly. In addition, we will add further results to discuss in more detail the dynamics of the buffet cycle (as shown in Fig. 2). Overall, we want to highlight the distinct features and relevance of the possibility of transonic buffet on wind turbine airfoils. We believe that these modifications will better substantiate the novelty and relevance of our work, such that the (revised) manuscript can be considered as suitable for publication in Wind Energy Science.

*1. The experiments are conducted at Reynolds numbers ($\sim 1 \times 10^6$) that are significantly lower than those in real turbine operations ($\sim 1 \times 10^7$). A more in-depth discussion of how this lower Reynolds number affects the results obtained would strengthen the interpretation of the results and help assess their applicability to real-world conditions.*

As pointed out in the manuscript and by the reviewer, the inflow Mach ($Ma_\infty$) and Reynolds ($Re$) numbers used in the current experimental study differ indeed from those of the IEA 15MW and 22MW reference wind turbines (RWT), see Gaertner et al. (2020) and Zahle et al. (2024), respectively. This choice is deliberate, with the purpose to reproduce qualitatively the same key physics (shock waves) as expected to occur on the full-scale wind turbine tip airfoil, while accounting for the limitations of the experimental facility as will be explained below.

[Figure]

(a) $\mathbf{Re} = 1.8 \times 10^6$          (c) $\mathbf{Re} = 9 \times 10^6$

Figure 4: Subsonic-supersonic boundary for the FFA-W3-211 wind turbine tip airfoil, with symbols indicating URANS simulations showing no supersonic flow (grey crosses), supersonic regime established (red circles), and configurations in which shock waves appear (green squares); from Vitulano et al. (2025a).

Perhaps counter to expectation, $Ma_\infty$ is not the single parameter relevant for the occurrence of compressibility effects, for a given angle of attack. Vitulano et al. (2025a) showed that an increase in $Re$ plays a crucial role in accelerating the appearance of shock waves at relatively lower $Ma$. This is illustrated in Fig. 4. For an $Re$ of $1.8 \times 10^6$, which is close to the $Re$ of ca. $1.6 \times 10^6$ achieved in the current experiments, shock waves occur only at a $Ma_\infty$ of 0.6. However, at an increased $Re$ of $9 \times 10^6$, i.e. at full scale, shock waves start to occur already at a $Ma_\infty$ of 0.45, close to an angle of attack of -9 deg. In addition, dynamic effects as a result of instantaneous angle of attack changes, e.g. due to turbulent gusts, can result in conditions where shock waves are observable at an even lower $Ma_\infty$ of ca. 0.35, i.e. close to inflow Mach numbers expected at the tip of the IEA 22MW RWT. The latter effect has been shown very recently in Vitulano et al. (2025b) using Unsteady Reynolds-Averaged Navier-Stokes (URANS) simulations of a pitching airfoil.

This combination of a very high $Re$, a relatively low $Ma_\infty$ and transonic flow over the airfoil is not possible to reproduce in most, if not all, experimental wind tunnel facilities available for wind energy research. However, we currently depend on experimental studies to foster the understanding of the transonic flow dynamics of thick wind turbine airfoils, in view of the limited capability of URANS-like numerical simulation tools to capture this. Therefore, to produce shock waves on such an airfoil experimentally, we selected a combination of $Ma_\infty$ and $Re$, such that the physics of shock occurrence under investigation in the experiments is qualitatively the same as expected for a full-scale turbine. We subsequently reduced $Ma_\infty$ at the same $Re$ to show how transonic flow without shocks can also be attained.

In the original manuscript, we briefly alluded to the argument made above when we mention the study by Vitulano et al. (2025a) on line 168. However, we agree with the reviewer that this needs to be discussed in more detail, in order to justify the selected operating conditions of the experiments. It is critical to understand that in order to reproduce the same physics of shock occurrence as in full-scale, the unavoidable reduction in $Re$, for any wind tunnel test, necessitates to adjust the accompanying $Ma_\infty$ and angle of attack (see Fig. 4). Therefore, in the revised manuscript, we intend to more clearly and explicitly justify the choice of $Ma_\infty$ and

*Re*, including additional available but unpublished CFD analyses and experimental data that corroborate the validity of the choices made with respect to $Ma_\infty$ and *Re*.

As per the discussion above, Experimental Design (section 2) in the revised manuscript has been extended (l. 82-107) to provide more clarity and justification of our choice of higher inflow Mach numbers. Fig. 4 in the current document is also included in the revised manuscript (as Fig. 2).

*2. The study presents only experimental data, without any supporting numerical simulations. This makes it difficult to validate or generalize the observations. While the experiments are of high quality, some comparison with CFD simulations would reinforce the conclusions. At present, the lack of numerical support limits the completeness of the study. This issue is reinforced by the fact that, as also commented on by the authors, the measurements shown in Figures 5 are in contrast to studies previously published by the same authors and summarized in Figure 4. Further analysis (through, for example, CFD simulations), might better explain this mismatch.*

In the work of Vitulano et al. (2025a), URANS simulations were utilised to explore a broad parameter space for the FFA-W3-211 airfoil, viz. $0.2 \leq Ma \leq 0.7$ and $1.8 \times 10^6 \leq Re \leq 9 \times 10^6$. This allowed to identify the conditions resulting in the occurrence of shock waves, which was not possible with a low-order tool like XFOIL, and subsequently helped inform the experimental design of the current study. However, the objectives of the experiments were different, namely to investigate and characterize the unsteady nature of the resulting shock waves, which could not be captured by the URANS simulations. For this reason, the authors consider that maintaining the focus on the experimental data for the current manuscript is justified. Furthermore, the experimental data will be made available, so that it can be used to validate/design numerical models for capturing shock waves in similar conditions.

A dedicated (conference) publication on validating the numerical setup used by Vitulano et al. (2025a) with our experiments is currently in preparation. Nevertheless, in line with the suggestions made by the reviewer to support conclusions and choices with simulations, we agree that including an additional section on the comparison between numerical and experimental results will form a valuable addition to the revised manuscript.

[Figure]

Figure 5: Normalized streamwise velocity fields for the FFA-W3-211 airfoil, for $Ma_\infty = 0.6$, AoA$= -6°$. Left: from numerical simulations (URANS, fully turbulent flow, $Re = 1.8$M); right: from experiments (tripped at 5% chord, $Re = 1.4$M).

A preview of the kind of material that will be added in the revision is shown in Fig. 5, which shows a comparison of normalized streamwise velocity fields for $Ma_\infty = 0.6$, AoA$= -10°$ obtained in simulations (left) and experiments (right). The results show a reasonable match in the overall flow physics. It is important to note that in Fig. 5, the URANS simulations were performed for a fully turbulent flow, and, to match this, the experimental data presented in this figure corresponds to a fixed-transition situation with the airfoil model tripped at 5% chord. On the other hand, the experiments presented in the current manuscript were conducted with a free-transition (untripped) airfoil model. However, the primary purpose of the data in Fig. 5 is to show that, for the same $Ma_\infty$ and a similar $Re$, URANS calculations capture the mean flow field of the experiments reasonably well. This gives some confidence in using the numerical approach in order to identify the transonic flow boundary and the occurrence of shocks, also for full-scale $Re$. To this end, more data will be shown in the revised manuscript as detailed below.

We plan for the following additions to compare experimental results with numerical simulations:

- An analysis of the transonic envelopes ($Ma_\infty$ vs AoA, as presented in Fig. 4 of the original manuscript) for clean (free-transition) and fixed-transition conditions using XFoil, in order to assess the impact of a

transitional boundary layer on the development of local supersonic flow. This is an expansion on Fig. 4 above, calculated with URANS using existing data (or new data that can be generated rather quickly).

- A qualitative comparison between the numerical and experimental velocity flow fields will be shown, to support that both the CFD simulations and experimental measurements capture the same physical phenomena, in a time-average sense. Plots similar to Fig. 5 are intended to be provided for both $Ma_\infty = 0.6$ and 0.5, with experiments tripped and in free-transition, and CFD simulations to compare. These plots are based on already available data.

In the revised manuscript, the following changes have been made:

- Figure 3 (Fig. 4 in the original manuscript) now includes XFoil calculations for both tripped and clean airfoils, as well as a transonic envelope developed through URANS simulations

[Figure]

Figure 6: Revised transonic envelope (Fig. 3 in the revised manuscript).

- In the Discussion (section 5), qualitative comparisons are made (l. 483-513) between the URANS simulations and experimentally measured mean velocity fields (Fig. 20 in the revised manuscript).

[Figure]

Figure 7: Mean normalized streamwise velocity fields from (left column) PIV measurements and (right column) URANS simulations for $Ma_\infty = 0.5$ (top row) and $Ma_\infty = 0.6$ (bottom row), at AoA= $-6$. A local Mach number of 1 is marked with a black dotted line. (Fig. 20 in revised manuscript).

*3. Abstract, l. 1 and 6 "cutout" while throughout the text cut-out is used.*

Thank you for pointing out the inconsistency. This will be taken care of in the revised manuscript. Only "cut-out" is used throughout the revised text.

*4. The Methodology section could benefit from including photos of the wind tunnel setup with the airfoil installed, if available.*

The following picture of the setup is included in the revised manuscript (Fig. 4).

[Figure]

Figure 8: Experimental setup (Fig. 4 in the revised manuscript)

*5. Table 1 is not referenced in the text; please clarify its relevance and refer to it explicitly.*

Thank you for noticing. This will be resolved in the revised manuscript.
In the revised manuscript, Table 2. "Sources of uncertainty" (Table 1 in the original manuscript) is referenced in the text. It lists the uncertainty involved in the measured/derived quantities (velocity, local Mach number, etc.) from various sources. Quantifying the uncertainties helps in building confidence in the measurements, given that the uncertainties are much smaller compared to the mean quantities (l. 183-186).

*6. Experimental Design, l. 169 "Furthermore, The"*

Corrected in the revised manuscript.

*7. Results, Local Mach Number Trends, l. 202 "(Figure 5f", missed )*

Corrected in the revised manuscript.

*8. Results, Local Mach Number Trends, l. 217 "(Fig. 6a", missed )*

Corrected in the revised manuscript.

*9. Results, Probability of Local Supersonic Flow, l.251 and 259 "Mainfty" appears to be a typo or LaTeX conversion error*

Corrected in the revised manuscript.

*10. In Figure 4, the figure caption tells " The transonic envelope showing the separation between complete subsonic flow and transonic flow for an FFA-W3-211 airfoil". but the graph indicates supersonic conditions with value of Mach less than 1. This could confuse the reader; consider clarifying the graph and/or caption.*

Figure 4 in the original manuscript depicts the transonic envelope (the boundary between a fully subsonic flowfield and a transonic flowfield, i.e. with local regions of supersonic flow), in terms of the angle-of-attack (AoA) and the free-stream Mach number ($Ma_\infty$) that the airfoil experiences. The latter is always subsonic ($Ma_\infty < 1$). The "supersonic" conditions refer to the occurrence of local regions of supersonic flow appearing in the flowfield around the airfoil due to flow acceleration on the suction side, which can happen even if the free-stream remains subsonic. This will be clarified in the revised manuscript.

In the revised manuscript, the revised figure caption (Fig. 3) clarifies the difference between the inflow Mach number ($Ma_\infty$) and having transonic flow, i.e, developing local pockets of supersonic flow over the airfoil.

*11. In Figure 7, consider thickening the dashed line to improve visibility against grayscale. Also, Figure 7d appears difficult to interpret and compare to the others—additional commentary explaining its significance would be helpful.*

For $Ma_\infty = 0.5$ and AoA= $-10°$, the mean (i.e., time-average) flow is not transonic (as seen in Figure 5e). Figure 7d shows the probability of supersonic flow over the airfoil for the same conditions as Figure 5e, highlighting the fact that $\sim 30\%$ of the time, local supersonic pockets occur near the leading edge. This was already discussed in the text (line 258) and will be elaborated on in the revised manuscript for clarity.

In the revised manuscript, Figure 8 (which was Figure 7 in the original manuscript) has been updated to include more information on the probability of flow separation along with supersonic flow (section 4.2). We hope the revised figure and discussion offer more clarity.

*12. Results, Occurrence of Shock Waves, Mach 0.55 is introduced without explanation. Given that earlier sections focus only on 0.5 and 0.6, please justify this intermediate condition.*

In Figure 8 of the original manuscript, schlieren images for $Ma_\infty = 0.55$ were added, alongside those for $Ma_\infty = 0.5$ and $Ma_\infty = 0.6$, to document Mach number effects on the flowfield, i.e. revealing the growing strength of the shocks with increasing Mach number. However, no PIV data was acquired for this intermediate case of $Ma_\infty = 0.55$, so it is not discussed elsewhere in the original manuscript. In the revised manuscript, we will further clarify the addition of this case to the schlieren analysis.

The intermediate condition ($Ma_\infty = 0.55$) has been removed from the main text. However, it has been included in Appendix B as supplementary results.

In summary, the decision to carry out the experiments in the wind tunnel at a higher Mach number, in combination with a lower Reynolds number than for full-scale, was made carefully and deliberately to balance the experimental limitations, with the goal to capture (qualitatively) the physics (of shock waves) that are expected to occur at full-scale. This will be explained in more detail in the revised manuscript. In addition, we will add available but previously unpublished data from URANS simulations to this discussion that were used to design the experiment and provide confidence in the relevance of our findings and our conclusions.

**Referee #3**

*This paper addresses a relatively unexplored aspect of transonic flows by investigating airfoils with high thickness-to-chord ratios. The reviewer appreciates this novel contribution. The experimental methodology is clearly described, and the results are well analyzed and documented.*

*The reviewer also acknowledges the authors' attempt to relate the findings to the possibility of transonic flow occurring at high negative angles of attack near the blade tip of a wind turbine. However, there are significant concerns regarding this interpretation. The primary issue is the influence of the open atmosphere. In the case of wind turbines, the surrounding unconfined air can respond to localized flow accelerations, allowing for upstream and downstream adjustments that may mitigate the extreme local accelerations observed in confined wind tunnel conditions. It is possible that, in an unconfined environment, the incoming flow may decelerate over a wider region, thereby reducing or preventing the occurrence of extreme suction peaks and transonic regions.*

*While the reviewer values the experimental investigation of thick airfoils under transonic conditions, extending these findings to system-level conclusions for wind turbines—based on low-fidelity tools—appears premature. For such generalizations to be justified, high-fidelity simulations of full-scale systems operating in open atmospheric conditions would be necessary.*

*In conclusion, the reviewer recommends that the scope of the manuscript be either limited to the experimental investigation of high-thickness airfoils under transonic conditions, which in itself provides valuable insights, or extended with comprehensive high-fidelity CFD studies to support the broader implications for wind turbine applications. Without such additions, system-level generalizations remain insufficiently supported.*

We sincerely thank the reviewer for the evaluation of our work and for recognizing the novelty of investigating transonic flow phenomena on thick airfoils. We also appreciate the constructive feedback regarding the interpretation of our results in the context of wind turbine operation.

We duly note the reviewer's concern that, in an unconfined atmospheric environment, the flow may adjust upstream and downstream, potentially mitigating the extreme accelerations observed in confined wind tunnel conditions. Furthermore, we acknowledge that our analysis is conducted at the 2D airfoil level, while the response of a real wind turbine blade tip is influenced by additional three-dimensional effects and the surrounding atmospheric flow.

To address this point, we will explicitly include a disclaimer in the conclusions of the revised manuscript, clarifying that our present study is limited to 2D thick airfoils under transonic conditions. While the results provide valuable insights into the fundamental aerodynamic behavior of such airfoils, system-level generalizations for wind turbines cannot be made without further high-fidelity 3D simulations in open atmospheric conditions.

We appreciate the reviewer's perspective and believe this clarification will improve the scope of our work.

In the revised manuscript, the conclusions (section 6) have been updated (l. 548-551) to emphasise that the current study is limited to 2D airfoil sections and that 3D effects require further scrutiny to extend the findings to the system-level.

**Associate Editor's report**

*1. Minor Comments*

*I have only one minor comment: when looking at Figures 2 and 3, which represent the subsonic–transonic envelope, an inconsistency appears between the two figures. These should represent the same envelope; however, in Figure 2 the upper region of the domain is labeled as supersonic regime, while in Figure 3 it is labeled as transonic regime. I suggest that the authors use consistent terminology to ensure clarity for the reader.*

Thank you to the reviewer for pointing this out. The final manuscript now has a revised Figure 2 (see below) as per the above suggestions.

[Figure]

Figure 9: Subsonic-transonic boundary for the FFA-W3-211 airfoil generated using XFoil (solid black line), with symbols representing URANS simulations showing only subsonic flow (grey crosses), local supersonic regime established (red circles), and configurations in which shock waves appear (green squares) for different Reynolds numbers; from Vitulano et al. (2025a). Included in the revised manuscript as Figure 2.

**References**

De Tavernier, D., & von Terzi, D. (2022). The emergence of supersonic flow on wind turbines. In *Journal of Physics: Conference Series* (pp. 042068).

D'Aguanno, A., Schrijer, F.F.J., & van Oudheusden, B.W. (2021). Experimental investigation of the transonic buffet cycle on a supercritical airfoil. *Experiments in Fluids*, 62, 1-23.

Giannelis, N.F., Vio, G.A., Levinski O. (2017). A review of recent developments in the understanding of transonic shock buffet. *Prog Aerosp Sci* 92:39–84.

Gaertner, E., Rinker, J., Sethuraman, L., Zahle, F., Anderson, B., Barter, G., Abbas, N., Meng, F., Bortolotti, P., Skrzypinski, W., Scott, G., Feil, R., Bredmose, H., Dykes, K., Shields, M., Allen, C., & Viselli, A. (2020). Definition of the IEA Wind 15-Megawatt Offshore Reference Wind Turbine. *Tech. Rep., National Renewable Energy Laboratory (NREL), Golden, CO.*

Zahle, F., Barlas, T., Lonbaek, K., Bortolotti, P., Zalkind, D., Wang, L., Labuschagne, C., Sethuraman, L., & Barter, G. (2024). Definition of the IEA Wind 22-Megawatt Offshore Reference Wind Turbine. *Tech. Rep., National Renewable Energy Laboratory (NREL), Golden, CO (United States).*

Vitulano, M., De Tavernier, D., De Stefano, G., & von Terzi, D. (2025a). Numerical analysis of transonic flow over the FFA-W3-211 wind turbine tip airfoil. *Wind Energy Science*, 10(1), 103–116.

Vitulano, M., De Tavernier, D., De Stefano, G., & von Terzi, D. (2025b). CFD analysis of dynamic wind turbine airfoil characteristics in transonic flow using URANS. *Wind Energ. Sci. Discuss.*, [preprint], https://doi.org/10.5194/wes-2025-125, in review.

von Terzi, D., De Tavernier, D., & Zaayer, M. (2023). Method of operating a wind turbine. *Patent WO2025127925A1*, international publication date 19.6.2025.